
# Validation of the Absorbing Aerosol Height Product from GOME-2 using CALIOP Aerosol Layer Information

Veerle De Bock[1], Alexander Mangold[1], L. Gijsbert Tilstra[2], Olaf N.E. Tuinder[2], Andy Delcloo[1]

[1]Royal Meteorological Institute of Belgium, Ringlaan 3, B-1180, Uccle, Belgium,
[2]Royal Netherlands Meteorological Institute, Utrechtseweg 297 , NL-3731 GA De Bilt, The Netherlands

*Correspondence to*: Veerle De Bock (veerledb@meteo.be)

**Abstract**

Within the framework of aviation safety, knowledge on the location and height of volcanic ash layers is of extreme importance. Several ground based instruments (such as lidars) can provide detailed information on the height and vertical
extent of these ash layers, however with a limited spatial coverage. The biggest advantage of satellite instruments is their ability to have near daily global coverage which makes them the perfect candidate for locating and tracking aerosol layers around the globe. Since the Global Ozone Monitoring Experiment 2 (GOME-2) instrument is carried on the MetOp series of operational satellites, it is designed to cover a long time period from 2007 until 2022 (and beyond) and global coverage is achieved within one day.

The GOME-2 Absorbing Aerosol Height (AAH) is a new product for aerosol detection, developed by the Royal Netherlands Meteorological Institute (KNMI) which uses the Absorbing Aerosol Index (AAI) to detect the presence of absorbing aerosol and derives the actual height of the absorbing aerosol layer in the O2-A band using the Fast Retrieval Scheme for Clouds from the Oxygen A band (FRESCO) algorithm. The first results of a quantitative validation of the AAH product focusing on case studies of volcanic eruptions will be presented here. For a total of 15 different volcanic eruptions, GOME-2 AAH data
are compared to the minimum and maximum aerosol layer height provided by Cloud-Aerosol Lidar with Orthogonal Polarization (CALIOP) for pixels within 100 km distance from each other. For GOME-2A and -2B, about 50 to 60% of the AAH pixels are within the EUMETSAT threshold requirements (for layers which are located lower than 10km, the maximum absolute difference should be within 3km; for layers which are located higher than 10km, the maximum absolute difference should be within 4km), while for GOME-2C this is about 70%. The optimal requirement threshold (for layers
which are located lower than 10km, the maximum absolute difference should be within 1km; for layers which are located higher than 10km, the maximum absolute difference should be within 2km) is reached for GOME-2A, GOME-2B and GOME-2C in 17%, 28% and 41.5% of the cases. If only tropospheric aerosol species are studied, the results improve. This can also be seen when looking at the mean error of GOME-2. GOME-2A, GOME-2B and GOME-2C are able to represent the minimum CALIOP layer height with a mean error of -2.5 ± 5km, -1.2 ± 5.9km and -2 ± 5.8km respectively. If the
stratospheric aerosol layers are removed from the data, the errors obtained are -0.2 ± 3.6km, -0.1 ± 5.4km and -0.8 ± 3.8km for GOME-2A, GOME-2B and GOME-2C respectively (for the minimum CALIOP layer height). The results from two





specific case studies (i.e. the Calbuco eruption in 2015 and the Sarychev Peak eruption in 2009) are highlighted and show that GOME-2 underestimates the height of volcanic ash layers. Especially if the layers are located at altitudes above 15 km, since GOME-2 is not able to detect these layers due to the loss of sensitivity of the FRESCO algorithm at these algorithms.

## 1 Introduction

Volcanoes are important sources of aerosol particles and gas phase precursors of secondary air pollutants that can have significant effects on solar and infrared radiation with consequences for atmospheric radiative forcing and climate (Le Treut et al., 2007). A major hazard from volcanic eruptions is volcanic ash. When present in the upper troposphere where jet aircraft fly, it can cause jet engine failure, damage to turbine blades and pitot static tubes, with the possibility of the loss of the aircraft and lives (Prata & Tupper, 2009). While it is highly important to detect ash clouds, it has proven to be a difficult task due to the unpredictable nature of volcanic activity. The Earth's wind circulation systems are able to spread ash and gas over large distances quickly. Apart from that, it is difficult to manage the aviation business where the economic cost of keeping aircraft on the ground or re-routing them must be considered.

The current operational regime for disseminating volcanic ash hazard warnings to aviation divides the world in nine regions, each with a Volcanic Ash Advisory Centre sited within a National Meteorological Service. It is important to have accurate knowledge about the height of volcanic ash layers in the atmosphere within the framework of aviation safety. Also, global monitoring is essential as ash particles can be transported over large distances away from their source. To that end, satellite observations can serve an important purpose as they can provide insight into global volcanism without inherent risk to equipment or life from eruptive products (Flower & Kahn, 2020). Volcanic plumes are routinely observed by a range of passive ultraviolet (UV), visible (VIS) and infrared (IR) measurements and active lidar observations (Flower & Kahn, 2020, WMO, 2015 and references therein). For instance, plume height and particle property constraints are provided by the Multi-angle Imaging SpectroRadiometer (MISR) and Cloud-Aerosol Lidar with Orthogonal Polarization (CALIOP) (Flower & Kahn, 2020). The Ozone Monitoring Instrument (OMI), Ozone Mapping and Profiler Suite (OMPS) and Sentinel-5p (S5p) are able to provide gaseous emission mapping and MODerate-resolution Imaging Spectroradiometer (MODIS) can provide surface thermal anomalies. Plume extent can be monitored by MODIS, MISR and other low earth orbit and geostationary instruments. The lidar measurements are the most accurate in measuring the aerosol plume height and are sensitive to small and diffuse particle layers. This however comes at the expense of spatial and temporal coverage. Polar satellites such as the MetOp series offer the advantage of (near) global and daily coverage and instruments such as the Global Ozone Monitoring Experiment (GOME)-2 have already been used for aerosol detection (Tuinder et al., 2019; Balis et al., 2016).

The Royal Netherlands Meteorological Institute (KNMI) developed the Absorbing Aerosol Index (AAI) product for GOME-2 (Tuinder et al., 2019) which indicates the presence of aerosols in the atmosphere. This AAI product is also an operational product of SCIAMACHY (Tilstra et al., 2012), GOME-1 (De Graaf et al., 2005) and OMI (Torres et al., 2007). The Aerosol Index consists of a signal contributed by aerosols that absorb UV/VIS radiation (AAI) and by aerosols that absorb only little





or no radiation (SCI = SCattering Index). The advantage of this product is that it is not sensitive to surface type and that it
can be defined in the presence of clouds, which is where most aerosol retrieval algorithms have problems. The aerosol types
most clearly seen with the AAI are desert dust, biomass burning and volcanic ash. The AAI is most sensitive to the Aerosol
Optical Depth (AOD) and the aerosol layer height. Generally, thick and/or high altitude aerosol layers produce larger AAI
values than thin and/or lower altitude aerosol layers (Balis et al., 2016). The AAI however does not provide information on
the altitude of aerosol layers.

To answer the need for global coverage of the altitude of volcanic ash layers, KNMI developed a new GOME-2 product, the
Absorbing Aerosol Height (AAH). This product builds on the above described AAI product and derives the actual height of
absorbing aerosol layers in the O2-A band using the Fast Retrieval Scheme for Clouds from the Oxygen A band (FRESCO;
Wang et al., 2008, Wang et al., 2012) algorithm. First results of GOME-2A aerosol layer height validation were presented by
Balis et al. (2016). They compared the GOME-2A layer height information with layer heights retrieved by ground based
lidar measurements and showed that GOME-2A underestimates the aerosol layer height observed by the lidars. However,
their study was only based on the results of one volcanic eruption case (the Eyjafjallajökull eruption in 2010) and they
concluded that more dedicated validation campaigns were needed.

In this work, the new Absorbing Aerosol Height product from GOME-2 is validated against CALIOP data, focusing on case
studies during multiple volcanic eruptions. Only GOME-2 AAH pixels with an AAI value higher than 4 are included in the
validation study to ensure that there are enough absorbing aerosols in terms of absorption intensity (Tilstra et al., 2019b).
The GOME-2 AAH values are compared to the CALIOP aerosol layer height for pixels located within a maximum distance
of 100 km from each other. Sect. 2 describes the GOME-2 and CALIOP instrument and the method applied to validate
GOME-2 AAH using CALIOP vertical layer information. The results from this validation exercise will be presented and
discussed in Sect. 3 and 4.

**2 Method**

**2.1. GOME-2 and the Absorbing Aerosol Height**

**2.1.1 GOME-2 instrument information**

The GOME-2 instrument onboard the MetOp satellite platforms is a nadir looking and scanning UV-VIS spectrometer that
measures backscattered solar light (Munro et al., 2016). The instrument measures in a spectral range from 240 to 790 nm
with a spectral resolution of 0.26-0.51 nm. The MetOp satellites are flying in sun-synchronous orbits with equator crossing
times of approximately 09:30 local time (descending node) and a repeat cycle of 29 days. The default swath width of the
GOME-2 scan is 1920 km, which gives a nadir pixel size of 80 x 40 km and enables global coverage in about 1.5 days. The
current primary GOME-2B (and also GOME-2C) is operated in this mode, whereas the older GOME-2A instrument is



operated in a reduced swath with a swath width of 960km and nadir ground pixel size of 40 x 40 km. GOME-2C is in orbit

since the 7[th] of November 2018. A more detailed description of the instrument can be found in Munro et al. (2016).

### 2.1.2 GOME-2 Absorbing Aerosol Height

The GOME-2 Absorbing Aerosol Height (AAH) is a new product developed by KNMI within EUMETSAT's Atmospheric Composition Satellite Application Facility (AC SAF). This product builds on a previously developed product, the AAI (Tuinder et al., 2019) and derives the actual height of the absorbing aerosol layer in the O2-A band using the FRESCO

algorithm. The AAH is very sensitive to cloud contamination because aerosols and clouds can prove difficult to distinguish. Therefore, the AAH is computed for different FRESCO cloud fractions. FRESCO is able to determine the height of an absorbing aerosol layer in the absence of clouds, but under certain conditions also in the presence of clouds (Wang et al., 2012). More details can be found in the Product User Manual (Tilstra et al., 2019b) and Algorithm Theoretical Basis Document (Tilstra et al., 2019a).


In summary, the AAH algorithm retrieves the following parameters:

(1) CF: effective aerosol/cloud fraction

(2) CH: aerosol/cloud height

(3) SA: scene albedo

(4) SH: scene height.

Two different aerosol/cloud layer heights (CH and SH) are determined by the AAH algorithm. It is up to the algorithm to decide which of the two is the best candidate to represent the actual AAH. To determine whether CH or SH should be reported as the AAH, the algorithm distinguishes three situations (regimes) and the effective cloud fraction is used to check in which of these regimes the solution is likely to be found:

| Regime A: | $CF \leq 0.25$ | AAH = CH (high reliability) |
| Regime B: | $0.25 < CF < 0.75$ | AAH = max(SH,CH) (medium reliability) |
| Regime C: | $CF \geq 0.75$ | AAH = CH (low reliability) |

The above scheme is based on the results of the study presented in Wang et al. (2012). Regime A refers to the situation in which there is only a low degree of cloud cover or if the AOD is sufficiently large to compensate for the presence of a cloud layer below the aerosol layer. In this case the results reported in Wang et al. (2012) clearly show that CH is close to the real height of the aerosol layer in almost all cases. Exceptions are cases with low aerosol amounts, but these scenes were filtered out beforehand by demanding that the AAI must be higher than the threshold value of 4.0 index points. Regime C is the

situation of a thick cloud layer present in the scene. In this case, an aerosol layer is only retrieved successfully when the aerosol layer is sufficiently thick. According to the results presented in Wang et al. (2012), the best value for the AAH is that of the cloud height. In most cases, however, the AAH is severely underestimated. The reliability is therefore characterized as "low". Finally, regime B is an intermediate regime, and the best estimate is the highest value from cloud height and scene



height. The AAH found this way is likely to underestimate the AAH in some cases, and the reliability attributed to this
regime is "medium".

Due to the use of the O2-A band in the FRESCO algorithm, the retrieval is insensitive to the signal above 15 km, hence the
AAH is limited to a maximum value of 15 km (Wang et al., 2010).

The accuracy requirements for the AAH product, as defined in the Product Requirements Document (Hovila et al., 2019),
can be found in Table 1. The GOME-2 AAH product is available in Near-Real time (NRT) and offline processing, from the
Level-1 data generated from the GOME-2 instruments onboard the MetOp-A, MetOp-B and MetOp-C satellite platforms.

**Table 1. Accuracy requirements defined for the AAH product (from Hovila et al., 2019).**

|  | **Layer height < 10 km** | **Layer height >10 km** |
|---|---|---|
| **Threshold** | 3 km | 4 km |
| **Target** | 2 km | 3 km |
| **optimal** | 1 km | 2 km |

### 2.2. CALIOP and the Vertical Feature Mask product

The Cloud-Aerosol Lidar with Orthogonal Polarization (CALIOP) is a two-wavelength polarization lidar that performs
global profiling of aerosols and clouds in the troposphere and lower stratosphere (Winker et al., 2009). CALIOP is the
primary instrument on the Cloud-Aerosol Lidar and Infrared Pathfinder Satellite Observations (CALIPSO) satellite, which
has flown in formation with the National Aeronautics and Space Administration (NASA) A-train constellation of satellites
since May 2006. The satellites of the A-train are in a 705 km sun-synchronous polar orbit, giving a 16-day repeat cycle, with
an equator-crossing time of about 13:30 local solar time. The orbit inclination of 98.2° provides global coverage from
CALIPSO between 82°N and 82°S. The spatial resolution from CALIOP data can be found in Table 2.

**Table 2: Spatial resolution of downlinked data from CALIOP (from Winker et al., 2009).**

| **Altitude range** | **Horizontal resolution** | **532 nm vertical resolution** | **1064 nm vertical resolution** |
|---|---|---|---|
| **(km)** | **(km)** | **(m)** | **(m)** |
| 30.1-40.0 | 5.00 | 300 | --- |
| 20.2-30.1 | 1.67 | 180 | 180 |
| 8.2-20.2 | 1.00 | 60 | 60 |
| -0.5 to 8.2 | 0.33 | 30 | 60 |
| -2.0 to -0.5 | 0.33 | 300 | 300 |



More detailed information on the satellite and instrument can be found in Winker et al. (2009). Data products derived from

the CALIOP measurements are distributed worldwide from the Atmospheric Science Data Center (ASDC) located at the

NASA Langley Research Center (LaRC) (Kim et al., 2018).

A suite of algorithms has been developed to identify aerosol and cloud layers and to retrieve a variety of optical and microphysical properties. The Scene Classification Algorithm (SCA) consists of a set of algorithms that perform typing of the detected layers based on layer height and layer-integrated properties. If the layer is classified as aerosol, SCA uses a

decision tree to classify the aerosol type. "Type" stands for a mixture of aerosol components that is characteristic of a region or an air mass. The mixture observed at a given location depends on local aerosol sources, wind trajectories and remote sources of aerosol, the state of internal and external mixing, chemical transformation processes that may have occurred during transport, and the state of hydration.

In this study, CALIOP version 4.20 Vertical Feature Mask (VFM) data are used. This version of data allows for the detection

of stratospheric aerosol layers (which was not possible in previous versions; Kim et al., 2018). The stratospheric aerosol subtyping algorithm performs well at identifying volcanic ash and sulfate above the tropopause (Kim et al., 2018). Note that below the tropopause, ash and sulfate plumes are given tropospheric aerosol subtypes: volcanic ash is often classified as dust or polluted dust and volcanic sulfate is often classified as elevated smoke (CALIPSO Users Guide at https://www-calipso.larc.nasa.gov/resources/calipso_users_guide/qs/cal_lid_l2_all_v4-20.php). As a result, contiguous aerosol features

crossing the tropopause will have aerosol subtypes which switch from tropospheric to stratospheric subtypes, depending on the relationship between the attenuated backscatter centroid altitude of the layer identified by the feature finder and the tropopause altitude. Weakly scattering stratospheric aerosol layers which are not classified as polar stratospheric aerosol are classified as "sulfate/other". Therefore, layers that are, in fact, ash and/or smoke could be misclassified as "sulfate/other" if they are weakly scattering (layer integrated attenuated backscatter less than 0.001 sr$^{-1}$).

The VFM product provides the latitude and longitude of the laser footprint (at the temporal midpoint of a 15 shot average for each 5 km layer of the feature classification flag data), the profile time, a day/night flag, a land/water flag and a feature classification flag. This feature classification flag is stored as a 16 bit integer and provides an assessment of

(a) the feature type: e.g. cloud vs aerosol vs stratospheric layer

(b) the feature subtype (see Table 3)

(c) layer ice-water phase

(d) amount of horizontal averaging required for layer detection.

(e) type and subtype quality assurance flag

In this study, only data which are defined by the CALIOP retrieval data as tropospheric or stratospheric aerosol are used and cloud layers are excluded from our analysis.




**Table 3: Feature type and subtypes in the CALIOP V4.20 VFM data.**

| Tropospheric aerosol | Stratospheric aerosol | Cloud |
|---|---|---|
| 0 = not determined | 0 = not determined | 0 = low overcast, transparent |
| 1 = clean marine | 1 = PSC aerosol | 1 = low overcast, opaque |
| 2 = dust | 2 = volcanic ash | 2 = transition stratocumulus |
| 3 = polluted continental/smoke | 3 = sulfate/other | 3 = low, broken cumulus |
| 4 = clean continental | 4 = elevated smoke | 4 = altocumulus (transparent) |
| 5 = polluted dust | | 5 = altostratus (opaque) |
| 6 = elevated smoke | | 6 = cirrus (transparent) |
| 7 = dusty marine | | 7 = deep convective (opaque) |

## 2.3. Validation method

Balis et al. (2016) already stated that the AAH product can be used to monitor volcanic eruptions globally and provide the height of the ash layers. So, within the framework of aviation safety, we decided to focus this validation study on a selection of case studies of confirmed volcanic eruptions. A list of confirmed volcanic eruptions can be found on the website of the Global Volcanism Program of the Smithsonian Institution (https://volcano.si.edu). From this list, cases with a 'clear' signal (AAI>4; Tilstra et al., 2019b) on AAI maps (available from the Tropospheric Emission Monitoring

Internet Service (TEMIS) website: http://www.temis.nl/airpollution/absaai) were chosen to further investigate. The final list of studied volcanic eruptions can be found in Table 4. GOME-2 AAH and CALIOP aerosol layer height were compared when the distance between the center pixel of GOME-2 and CALIOP was less than or equal to 100 km to maximize the probability of both instruments observing the same aerosol layer. There was no threshold used to limit the time difference between both satellite overpasses. Only AAH data with AAI>4 are validated to assure that there are enough absorbing

aerosols in terms of absorption intensity (Tilstra et al., 2019b). Using this approach, one GOME-2 AAH pixel can be compared to different CALIOP overpasses and also to different CALIOP vertical layers at the same location.

**Table 4: List of studied volcanoes with their geographical coordinates, the study period and the available GOME-2 overpasses.**

| Volcano (Country) | Latitude/Longitude | Studied dates | GOME-2 | # of data pairs |
|---|---|---|---|---|
| Barren Island (India) | 12.278°N/93.858°E | 25-26 September 2018 | B | B: 479 |
| Bulusan (Philipines) | 12.769°N/124.056°E | 3 May 2015 | B | B: 75 |
| Calbuco (Chile) | 41.33°S/72.62°W | 23-24 April 2015 | A/B | A: 1995 <br> B: 2961 |
| Eyjafjallajokull (Iceland) | 63.633°N/19.633°W | 15-16 April 2010 | A | A: 123 |





| Grimsvotn (Iceland) | 64.416°N/17.316°W | 23 May 2011 | A | A: 222 |
|---|---|---|---|---|
| Kasatochi (United States) | 52.177°N/175.508°W | 8 August 2008 | A | A: 909 |
| Krakatau (Indonesia) | 6.102°S/105.423°E | 22 February 2017 | B | B: 144 |
| Mount Kelud (Indonesia) | 7.93°S/112.31°E | 14-15 February 2014 | B | B: 216 |
| Nishinoshima (Japan) | 27.247°N/140.874°E | 13-14 July 2018 | A/B | A: 53 B: 413 |
| Paluweh (Indonesia) | 8.32°S/121.71°E | 2-11 February 2013 | A/B | A: 300 B: 231 |
| Puyehue-Cordon Caulle (Chili) | 40.59°S/72.12°W | 5 June 2011 | A | A: 72 |
| Raikoke (Russia) | 48.29°N/153.25°E | 22-23 June 2019 | B/C | B: 934 C: 427 |
| Rinjani (Indonesia) | 8.42°S/116.47°E | 26-27 October 2015 | A/B | A: 1405 B: 1240 |
| Sarychev Peak (Russia) | 48.092°N/153.20°E | 14-17 June 2009 | A | A: 543 |
| Ubinas (Peru) | 16.355°S/70.903°W | 13-14 September 2016 | A/B | A: 363 B: 12 |

## 3 Results

### 3.1 General results

All studied cases are analyzed together and it is determined how well GOME-2A, GOME-2B and GOME-2C perform for these specific cases by taking into account the accuracy requirements defined in Table 1. Results from this exercise are shown in Table 5 and Fig. 1.

Figure 1 limits the minimum CALIOP layer height to 15 km as this represents the detection limit for the GOME-2 AAH algorithm. It must be mentioned that the datasets for the three GOME-2 instruments are not the same (i.e. different volcanic cases are considered for each instrument) so caution must be taken when comparing them. Also, the dataset from GOME-2C is noticeably smaller than those for the other two instruments, due to its shorter time in orbit. For GOME-2A, -2B and -2C, the amount of compared pixel pairs is 5985, 6705 and 427 respectively. A list of used data is given in Table 4.

Overall, just about 50-60% of the AAH pixels from GOME-2A and -2B reach the threshold requirements. For GOME-2C, the number is higher, around 70%. The optimal requirement threshold is reached for GOME-2A, GOME-2B and GOME-2C in 17%, 28% and 41.5% of the cases (when comparing the AAH with the minimum CALIOP layer height). If only the tropospheric aerosol species (as defined by CALIOP) are studied, the results improve. This can also be seen from Table 5 (values between brackets). The accuracy requirement analysis was also performed per aerosol type (as defined by CALIOP). The results and discussion can be found the supplement (Tables S1, S2 and S3)



**Table 5: Percentage of data for each GOME-2 instrument that reached the threshold, target and optimal requirements. Results are shown for GOME-2 AAH minus minimum CALIOP layer height (AAH-minC) and for GOME-2 AAH minus the maximum CALIOP layer height (AAH-maxC). Values obtained when only considering the tropospheric aerosol species are shown between brackets.**

**GOME-2A**

|  |  | Layer height <10 km | Layer height >10 km | Total |
|---|---|---|---|---|
| **Threshold** | **AAH-minC** | 57.8 % (74.0 %) | 59.7 % (87.5 %) | 58.0 % (74.1 %) |
|  | **AAH-maxC** | 57.4 % (73.5 %) | 29.4 % (87.5 %) | 54.7 % (73.5 %) |
| **Target** | **AAH-minC** | 40.2 % (51.5 %) | 28.5 % (87.5 %) | 39.1 % (51.5 %) |
|  | **AAH-maxC** | 38.2 % (48.9 %) | 11.7 % (87.5 %) | 35.6 % (49 %) |
| **Optimal** | **AAH-minC** | 17.8 % (22.9 %) | 12.6 % (75 %) | 17.3 % (22.9 %) |
|  | **AAH-maxC** | 18.1 % (23.1 %) | 3.0 % (87.5 %) | 16.6 % (23.2 %) |

**GOME-2B**

|  |  | Layer height <10 km | Layer height >10 km | Total |
|---|---|---|---|---|
| **Threshold** | **AAH-minC** | 50.8 % (56.8 %) | 71.1 % (11.4 %) | 53.2 % (54.9 %) |
|  | **AAH-maxC** | 48.6 % (54.4 %) | 49.8 % (11.4 %) | 48.8 % (52.5 %) |
| **Target** | **AAH-minC** | 43.6 % (48.8 %) | 53.0 % (4.6 %) | 44.7 % (46.8 %) |
|  | **AAH-maxC** | 34.6 % (38.7 %) | 44.1 % (11.4 %) | 44.7 % (37.5 %) |
| **Optimal** | **AAH-minC** | 28.2 % (31.5 %) | 29.3 % (4.6 %) | 28.3 % (30.3 %) |
|  | **AAH-maxC** | 19.3 % (21.6 %) | 20.4 % (4.6 %) | 19.4 % (20.8 %) |

**GOME-2C**

|  |  | Layer height <10 km | Layer height >10 km | Total |
|---|---|---|---|---|
| **Threshold** | **AAH-minC** | 70.0 % (74.9 %) | No data (No data) | 70.0 % (74.9 %) |
|  | **AAH-maxC** | 72.4 % (77.4 %) | No data (No data) | 72.4 % (77.4 %) |
| **Target** | **AAH-minC** | 53.4 % (57.1 %) | No data (No data) | 53.4 % (57.1 %) |
|  | **AAH-maxC** | 64.9 % (69.4 %) | No data (No data) | 64.9 % ( 69.4 %) |
| **Optimal** | **AAH-minC** | 41.5 % (44.4 %) | No data (No data) | 41.5 % (44.4 %) |
|  | **AAH-maxC** | 43.3 % (46.4 %) | No data (No data) | 43.3 % (46.4 %) |






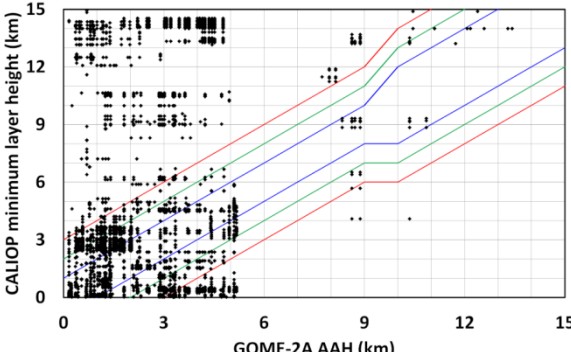

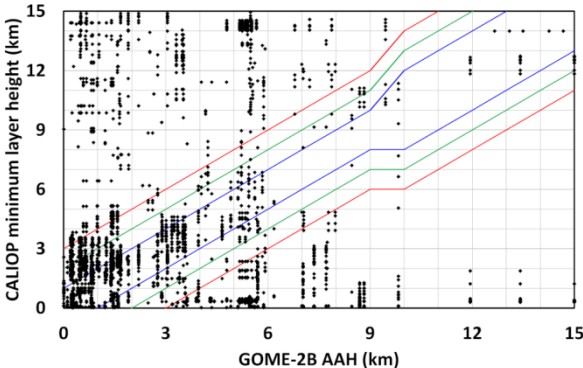

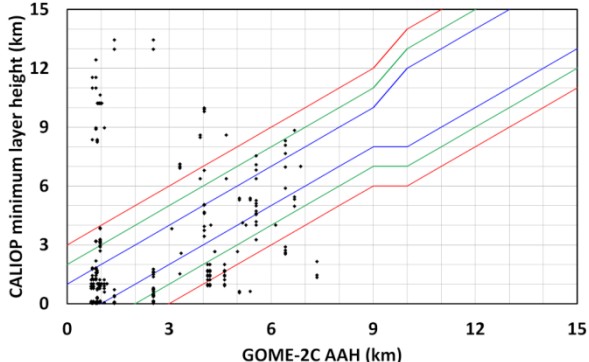

**Figure 1: Requirement plots for GOME-2A (upper left), GOME-2B (upper right) and GOME-2C (lower middle). The red, green and blue line represent the threshold, target and optimal requirement lines. CALIOP pixels are only shown up to a height of 15 km, which is the detection limit of GOME-2.**

In Fig. 1 there is a cloud of points for which the CALIOP layer height is between 12-15 km and the corresponding GOME-2 AAH is much lower (< 5 km) (especially for GOME-2A and GOME-2B). For GOME-2A, most of these pixels (85 %) were classified as volcanic ash, sulfate or elevated smoke layers and are classified by GOME-2 as pixels with high reliability. For GOME-2B however, only 28% of the pixels were classified as stratospheric aerosol species by CALIOP but 95% of the pixels have a medium or low reliability level.

For the selected case studies, all GOME-2 and CALIOP pixels within a 100 km distance range were compared (AAH versus minimum (minC) and maximum (maxC) CALIOP layer height). The results (height difference as a function of the distance) are shown in Fig. 2 for GOME-2A, GOME-2B and GOME-2C. Similarly, the height differences as a function of difference in overpass time for GOME-2A, GOME-2B and GOME-2C are presented in Fig. 3. (In both Fig. 2 and Fig. 3, results also include comparisons with CALIOP layers higher than 15 km.) From Fig. 2 and Fig. 3, it can be concluded that for all three

GOME-2 instruments, there is a large spread in the difference between the AAH and the CALIOP layer heights and there is no clear relation in function of the distance or time difference between overpasses. It needs to be specified that care must to





be taken in comparing the three instruments as the plots are not based on the same data for each instrument. E.g. for GOME-2C, only data from the Raikoke eruption have been used.

The overall performance of the three GOME-2 instruments is shown in Table 6 and is expressed by the mean and standard
deviation of the difference between GOME-2 AAH and CALIOP layer height.

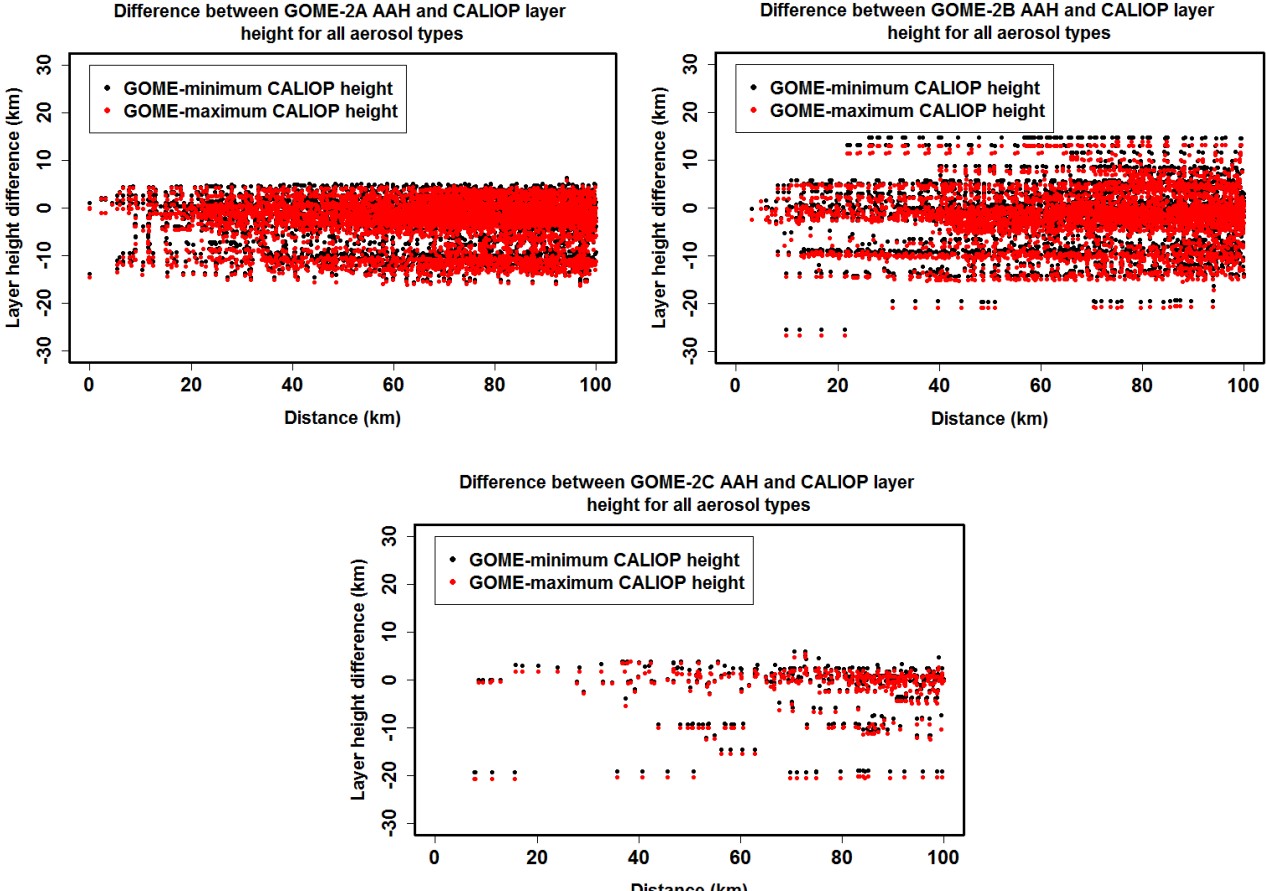

**Figure 2: Difference between GOME-2 AAH and the minimum (in black) and maximum (in red) CALIOP layer height in function of the distance between the GOME-2 and CALIOP pixel. The upper left plot shows the results for GOME-2A, the upper right plot**
**shows the results for GOME-2B and the lower middle plot shows the results for GOME-2C.**



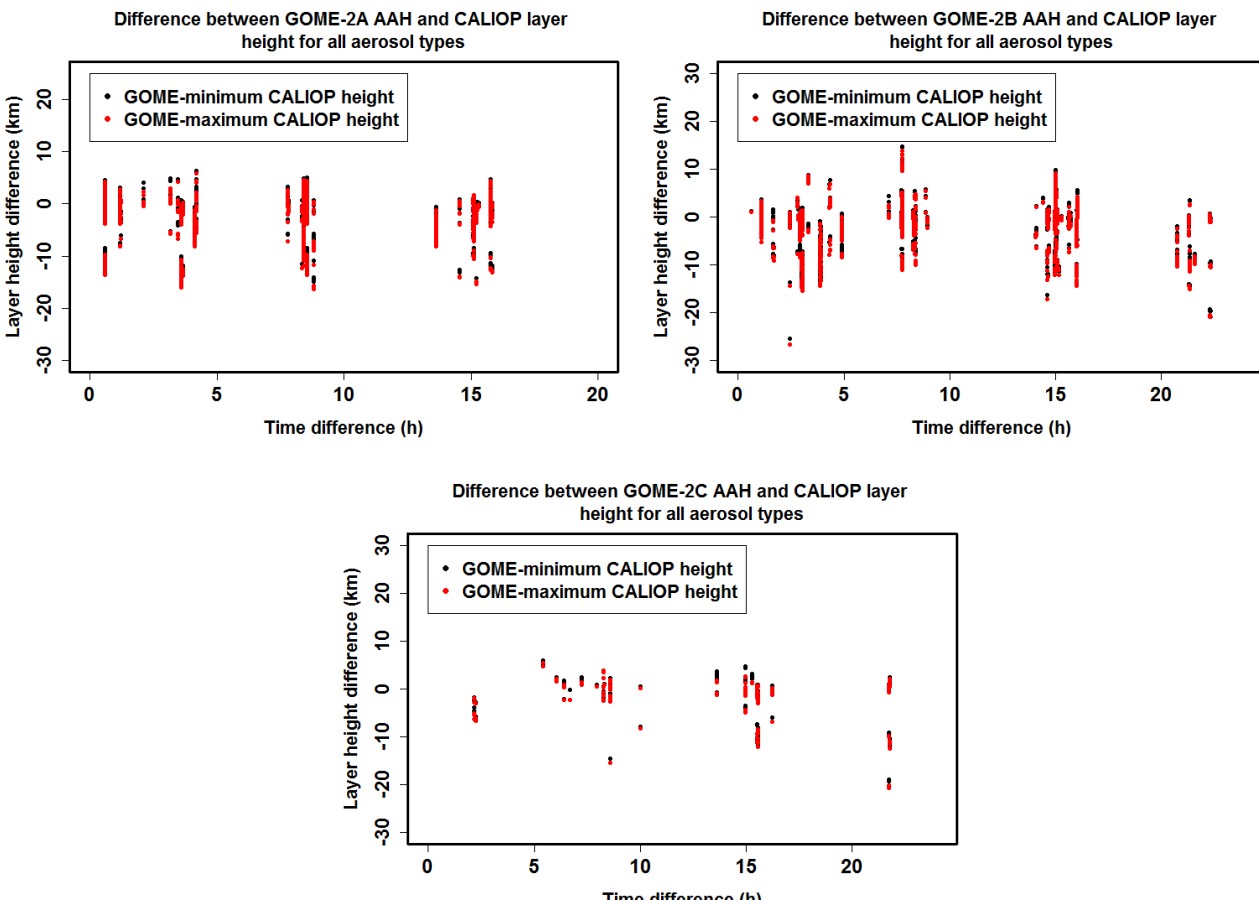

**Figure 3: Difference between GOME-2 AAH and the minimum (in black) and maximum (in red) CALIOP layer height in function of the time difference between overpasses from GOME-2 and CALIOP. The upper left plot shows the results for GOME-2A, the upper right plot shows the results for GOME-2B and the lower middle plot shows the results for GOME-2C.**

**Table 6: Overview of the mean difference (mean) and its standard deviation (stdev) between GOME-2 AAH and CALIOP minimum and maximum layer height.**

| | GOME-2A | | | |
|---|---|---|---|---|
| | min_dif (km) | | max_dif (km) | |
| **TYPE** | **mean** | **stdev** | **mean** | **stdev** |
| **ALL** | -2.5 | 5 | -3.3 | 5.1 |
| **Only tropospheric** | -0.2 | 3.6 | -1.0 | 3.6 |
| **clean marine** | 0.8 | 1.6 | -0.1 | 1.7 |
| **dust** | -1.4 | 4.3 | -2.3 | 4.2 |
| **polluted continental** | 1.1 | 2 | 0.1 | 1.8 |



| | min_dif (km) | | max_dif (km) | |
|---|---|---|---|---|
| clean continental | -1.1 | 1.4 | -1.5 | 1.6 |
| polluted dust | 1.4 | 2.7 | 0.8 | 2.9 |
| smoke | -1.4 | 1.6 | -2.3 | 1.4 |
| dusty marine | -0.8 | 2 | -1.6 | 2 |
| volcanic ash | -7.9 | 3.5 | -8.9 | 3.4 |
| sulfate | -8.3 | 1.6 | -9.8 | 1.3 |
| elevated smoke | -8.2 | 3.7 | -8.7 | 3.7 |

| GOME-2B | | | | |
|---|---|---|---|---|
| | min_dif (km) | | max_dif (km) | |
| TYPE | mean | stdev | mean | stdev |
| ALL | -1.2 | 5.9 | -2.1 | 5.9 |
| Only tropospheric | -0.1 | 5.4 | -1.0 | 5.4 |
| clean marine | 3.9 | 3.4 | 3.1 | 3.3 |
| dust | -3.1 | 5.4 | -4 | 5.5 |
| polluted continental | 3.8 | 5.3 | 2.9 | 5.1 |
| clean continental | -0.8 | 1.1 | -1.3 | 0.9 |
| polluted dust | 0.9 | 6.1 | 0.2 | 5.9 |
| smoke | -1 | 2.4 | -2 | 2.6 |
| dusty marine | 1.7 | 3.3 | 0.8 | 3.6 |
| volcanic ash | -5.9 | 4.8 | -6.9 | 4.8 |
| sulfate | -9.7 | 7 | -10.5 | 6.9 |
| elevated smoke | -5.5 | 5.1 | -6.2 | 5 |

| GOME-2C | | | | |
|---|---|---|---|---|
| | min_dif (km) | | max_dif (km) | |
| TYPE | mean | stdev | mean | stdev |
| ALL | -2 | 5.8 | -2.6 | 5.9 |
| Only tropospheric | -0.8 | 3.8 | -1.4 | 3.9 |
| clean marine | 1.5 | 1 | 0.9 | 1.1 |
| dust | -2.3 | 4.5 | -2.9 | 4.7 |
| polluted continental | 1.1 | 1.5 | 0.3 | 1 |
| clean continental | -0.8 | 0.1 | -1.1 | 0.2 |
| polluted dust | 0 | 3.5 | 0.8 | 3.5 |
| smoke | -0.7 | 2.2 | -1.3 | 2.2 |





| | | | | |
|---|---|---|---|---|
| **dusty marine** | 0.5 | 0.5 | 0.3 | 0.6 |
| **volcanic ash** | -19.2 | 0.1 | -20.5 | 0.1 |
| **sulfate** | -14.6 | 0 | -15.4 | 0 |
| **elevated smoke** | NO DATA | NO DATA | NO DATA | NO DATA |

GOME-2A, GOME-2B and GOME-2C are able to represent the minimum CALIOP layer height with a mean error of -2.5 ± 5 km, -1.2 ± 5.9 km and -2.0 ± 5.8 km respectively. For the maximum CALIOP layer height, the mean errors are -3.3 ± 5.1 km, -2.1 ± 5.9 km and -2.6 ± 5.9 km respectively. The high standard deviation is due to the inclusion of stratospheric aerosol species. It is clear that for the 'stratospheric' aerosol types (volcanic ash, sulfate and elevated smoke), the AAH is in most cases not able to represent the height of these layers. Layers higher than 15 km cannot be detected by GOME-2 as the

FRESCO algorithm is currently not sensitive at these altitude levels. The performance for the tropospheric aerosol subtypes is much better. If the stratospheric aerosol layers are removed from the data, the errors become -0.2 ± 3.6 km, -0.1 ± 5.4 km and -0.8 ± 3.8 km for GOME-2A, GOME-2B and GOME-2C respectively for the minimum CALIOP layer height and -1.0 ± 3.6 km, -1.0 ± 5.4 km and -1.4 ± 3.9 km for GOME-2A, GOME-2B and GOME-2C respectively for the maximum CALIOP layer height. Only the height of dust layers seems to be problematic. The height of the other species is approximated by

GOME-2A to within ~5 km. For GOME-2B, the differences tend to be a bit higher, but as not exactly the same dataset was used, it could be due to the contents of the dataset.

Table 7 shows the mean and standard deviation of the difference between GOME-2 AAH and CALIOP layer height for the different reliability levels (high, medium and low reliability level; as defined in Sect. 2.1.2). This study shows that most AAH pixels in this study are classified as having medium reliability (74%, 72% and 84% of the pixels for GOME-2A, -2B

and -2C respectively). For GOME-2A, the mean and standard deviation are lowest for the pixels with low reliability. A possible explanation can be found in the fact that for the low reliability GOME-2A dataset, the mean height of the CALIOP aerosol layers was clearly lower (around 3 km) than for the high and medium dataset (with a mean CALIOP layer height of respectively 7 and 6 km). For GOME-2B and -2C, the performance of the high reliability pixels is better than for the other reliability levels. For each AAH pixel, the error on the AAH is also given. Figure 4 shows the AAH plus and minus this error

and the minimum and maximum CALIOP layer height in function of the GOME-2 AAH for all three instruments. In the plots, the height of the CALIOP layers is limited to 15 km, which is the detection limit of GOME-2 as a result of the application of the FRESCO algorithm. On average, the errors are quite small: 0.4 km, 0.4 km and 0.3 km for GOME-2A, -2B and -2C respectively. Figure 5 shows the boxplots of the differences between the AAH from each GOME-2 instrument and the minimum CALIOP layer height for the different aerosol types (as defined by CALIOP). All boxplot results need to be

analyzed with caution as they are based only on specific case studies. Especially in the case of GOME-2C, only a very limited amount of data was examined. However, even with only case studies, a clear difference can already be seen between the tropospheric and stratospheric aerosol species, with differences between GOME-2 AAH and CALIOP layer height clearly higher for volcanic ash, sulfate and elevated smoke. Within the tropospheric aerosol species, differences are also





obvious. Dust and polluted dust have a larger spread compared to aerosol types that typically can be found very close to the

surface (e.g. clean marine).

**Table 7: Overview of the mean difference (mean) and its standard deviation (stdev) between GOME-2 AAH and CALIOP minimum (mindif) and maximum (maxdif) layer height for the different reliability levels. Behind each reliability level, the available number of GOME-2 data points is given.**

| | GOME-2A | | | |
| --- | --- | --- | --- | --- |
| | min_dif (km) | | max_dif (km) | |
| | mean | stdev | mean | stdev |
| High reliability (#: 32) | -2.5 | 5.9 | -3.4 | 6.0 |
| Medium reliability (#: 112) | -2.5 | 4.3 | -3.4 | 4.4 |
| Low reliability (#: 8) | -1.3 | 3.1 | -2.3 | 3.1 |
| | GOME-2B | | | |
| | min_dif (km) | | max_dif (km) | |
| | mean | stdev | mean | stdev |
| High reliability (#: 26) | 0.8 | 6.4 | -0.2 | 6.2 |
| Medium reliability (#: 110) | -1.6 | 5.7 | -2.4 | 5.7 |
| Low reliability (#: 16) | -3.4 | 4.5 | -4.3 | 4.6 |
| | GOME-2C | | | |
| | min_dif (km) | | max_dif (km) | |
| | mean | stdev | mean | stdev |
| High reliability (#: 3) | -0.2 | 4.8 | -0.7 | 4.8 |
| Medium reliability (#: 37) | -2.2 | 6.1 | -2.9 | 6.3 |
| Low reliability (#: 4) | -1.7 | 2.7 | -2.2 | 2.6 |






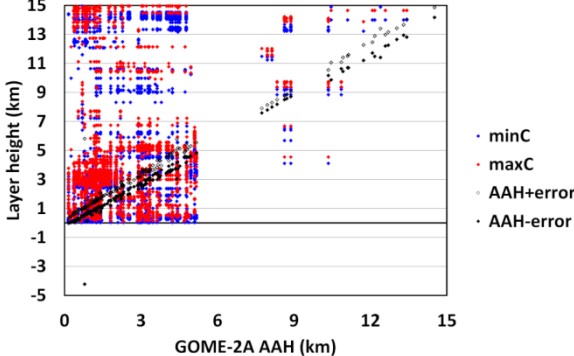
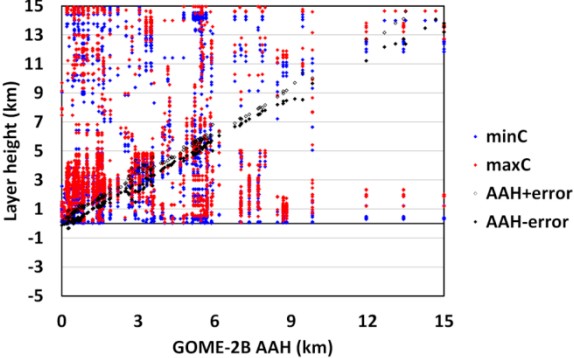

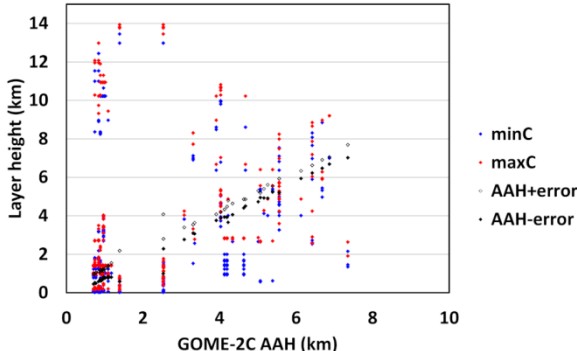

**Figure 4: GOME-2 AAH plus (in grey) and minus (in black) its error and the minimum (in blue) and maximum (in red) CALIOP layer height in function of GOME-2 AAH for GOME-2A (upper left), GOME-2B (upper right) and GOME-2C (lower middle). CALIOP pixels are only shown up to a height of 15 km, which is the detection limit of GOME-2.**



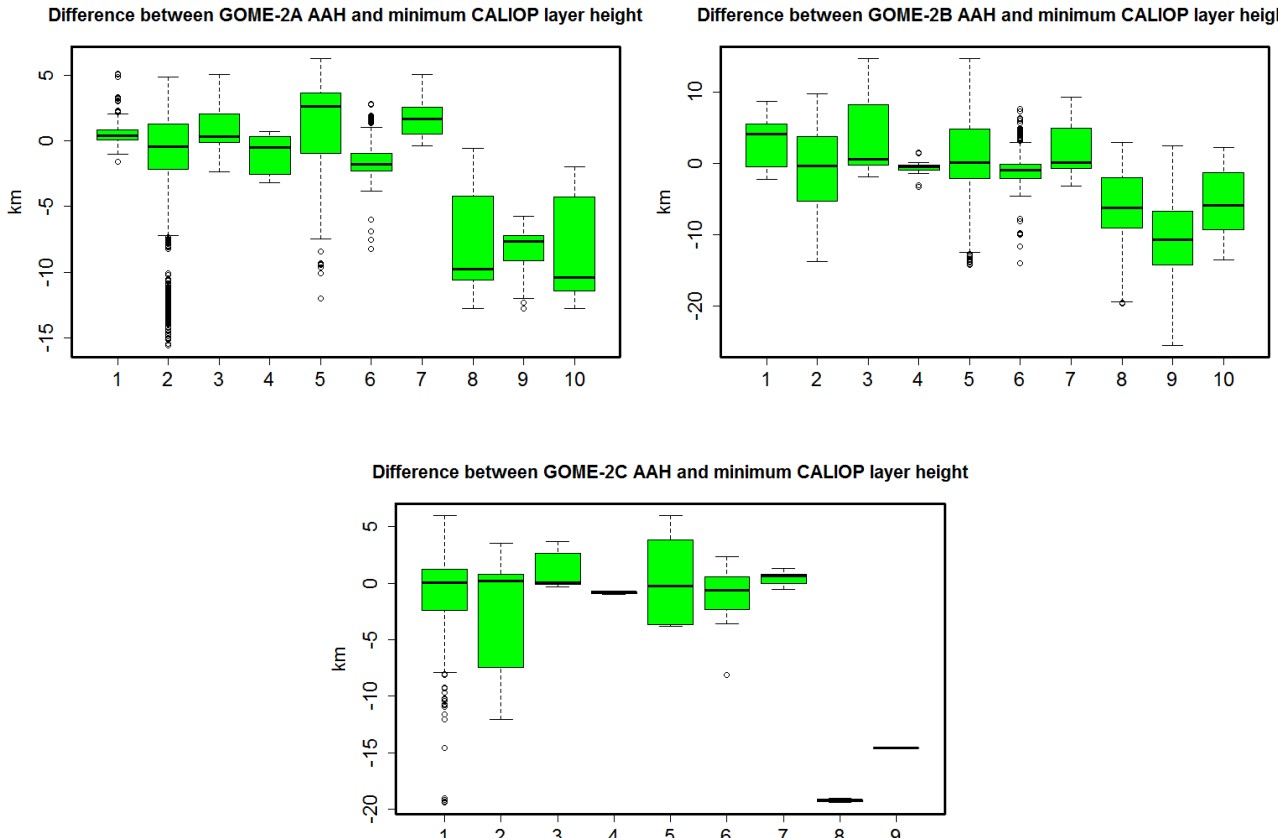

**Figure 5: Boxplot of differences between GOME-2A (upper left), GOME-2B (upper right) and GOME-2C (lower middle) AAH and CALIOP minimum layer height for the different aerosol species (1: clean marine; 2: dust; 3: polluted continental; 4: clean continental; 5: polluted continental; 6: smoke; 7: dusty marine; 8: volcanic ash; 9: sulfate; 10: elevated smoke)**

## 3.2 Case studies

The most important findings for a few of the volcanic case studies listed in Table 4 will be presented here. The focus will be on the eruption of the Calbuco volcano in 2015 and the Sarychev Peak eruption in 2009.

### 3.2.1 Calbuco eruption

The Servicio Nacional de Geología y Minería reported that an eruption from Calbuco occurred on the 23 April 2015 around 01h00, which lasted six hours and generated an ash plume that rose higher than 15 km and drifted towards the N, NE and E (Global Volcanism Program of the Smithsonian Institution, 2020). On the 24[th] of April 2015 the ash plume continued to rise 2 km and explosions were detected. Data from the 23[rd] until the 24[th] of April 2015 have been analyzed for both GOME-2A and GOME-2B.





**GOME-2A**

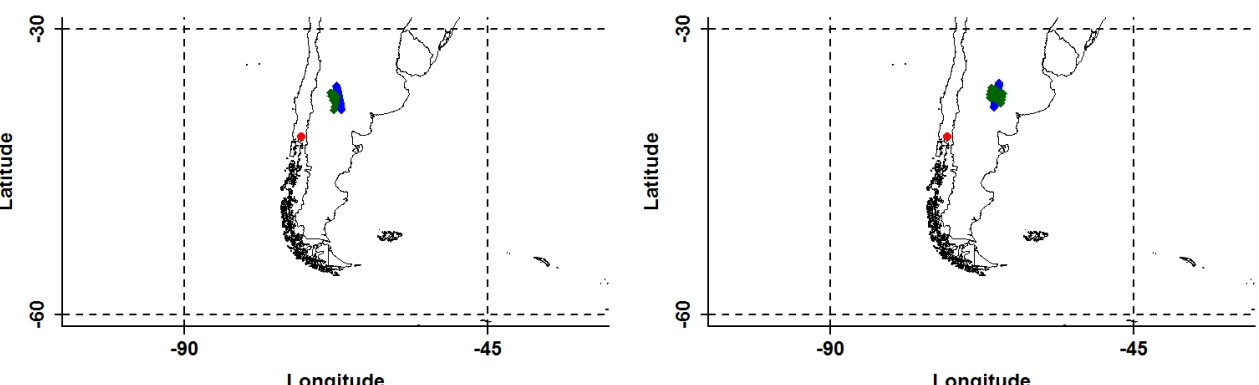

**Figure 6: The location of the volcano is shown in red. The location of the CALIOP and GOME-2A overpasses on the 23rd of April 2015 (left) and 24th of April 2015 (right) are shown in blue and green respectively.**

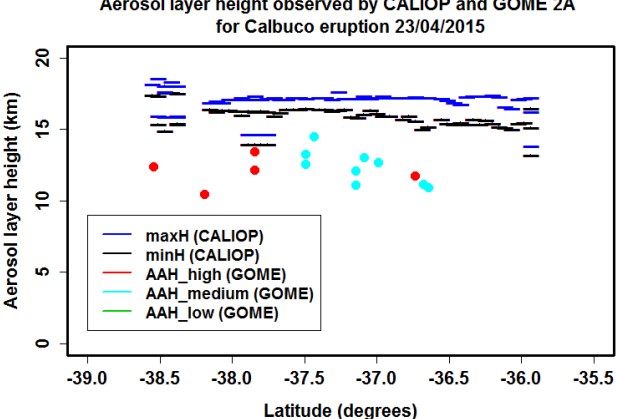


**Figure 7: The aerosol layer height detected by CALIOP is shown in blue (maximum layer height) and black (minimum layer height) as a function of latitude for the 23rd of April 2015. The GOME-2A AAH pixels located within 100 km of a CALIOP overpass pixel are presented in red (high reliability), cyan (medium reliability) and green (low reliability).**

Figure 6 shows the position of the GOME-2A and CALIOP overpasses near the Calbuco volcano on the 23[rd] and 24[th] of

April 2015. Figure 7 shows the observed aerosol layer height from CALIOP and the AAH detected by GOME-2A in

function of latitude for the 23[rd] of April 2015. All CALIOP pixels from Fig. 7 were classified as volcanic ash (with the

exception of one pixel classified as elevated smoke). The volcanic ash was detected by CALIOP between 13.2 and 18.6 km

while the AAH detected by GOME-2A for pixels within 100 km of CALIOP pixels was between 10.5-14.5 km (Table 8).

The time difference between the CALIOP and GOME-2A overpass was around 4 hours and the closest GOME-2A pixel was

located about 26 km from a CALIOP pixel. GOME-2A was not entirely able to capture the volcanic ash layer detected by

CALIOP as it was located at an altitude higher than 15 km.





On the 24ᵗʰ of April 2015, CALIOP detected volcanic ash at heights between 13.1-17.5 km, dust between 0.2-5.6 km, polluted dust between 0.2-5.5 km, sulfate and elevated smoke between 14.5-14.9 km (Table 8). Figure 8 shows the observed volcanic ash and dust layer height from CALIOP and the AAH detected by GOME-2A in function of latitude. The time

difference between both overpasses is around 8 hours and the closest GOME-2A pixel is located ~5 km from a CALIOP pixel. For this case it seems that GOME-2A does not see the volcanic species, but more likely the tropospheric dust and/or the polluted dust layer, detected by CALIOP.

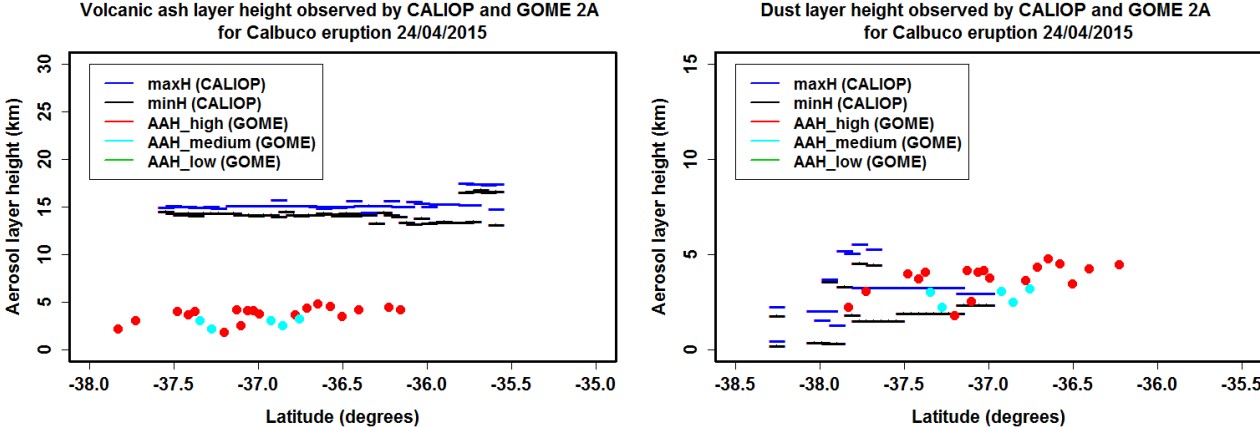

**Figure 8: Detail of the volcanic ash (left) and dust (right) layer detected by CALIOP. The maximum and minimum layer heights are shown in blue and black respectively as a function of latitude for the 24th of April 2015. The GOME-2A AAH pixels located within 100 km of a CALIOP overpass pixel are presented in red (high reliability), cyan (medium reliability) and green (low reliability).**

**Table 8: Height of the aerosol layer detected by CALIOP for the different aerosol species compared to the AAH detected by**
**GOME-2A for pixels located within 100 km distance for the 23rd and 24th of April 2015 (Calbuco eruption).**

|  | 23rd of April 2015 | | 24th of April 2015 | |
|---|---|---|---|---|
|  | CALIOP | AAH | CALIOP | AAH |
| **Dust** | Not detected | Not detected | 0.2-5.6 km | 1.8-4.8 km |
| **Polluted dust** | Not detected | Not detected | 0.2-5.5 km | 1.8-4.8 km |
| **Volcanic ash** | 13.2-18.6 km | 10.5-14.5 km | 13.1-17.5 km | 1.8-4.8 km |
| **sulfate** | Not detected | Not detected | 14.5-14.9 km | 1.8-4.2 km |
| **Elevated smoke** | 15.4-16.9 km | 10.5-13.4 km | 14.5-14.9 km | 1.8-4.8 km |

**GOME-2B**

Figure 9 shows the location of the overpass pixels of GOME-2B and CALIOP located within 100 km from each other for the 23ʳᵈ and 24ᵗʰ of April 2015. Figure 10 shows the observed aerosol layer height from CALIOP and the AAH detected by



GOME-2B in function of latitude for the 23rd of April 2015. All CALIOP pixels from Fig. 10 were classified as volcanic ash (with the exception of one pixel of elevated smoke). The AAH detected by GOME-2B was between 9.1-14.7 km (Table 9).

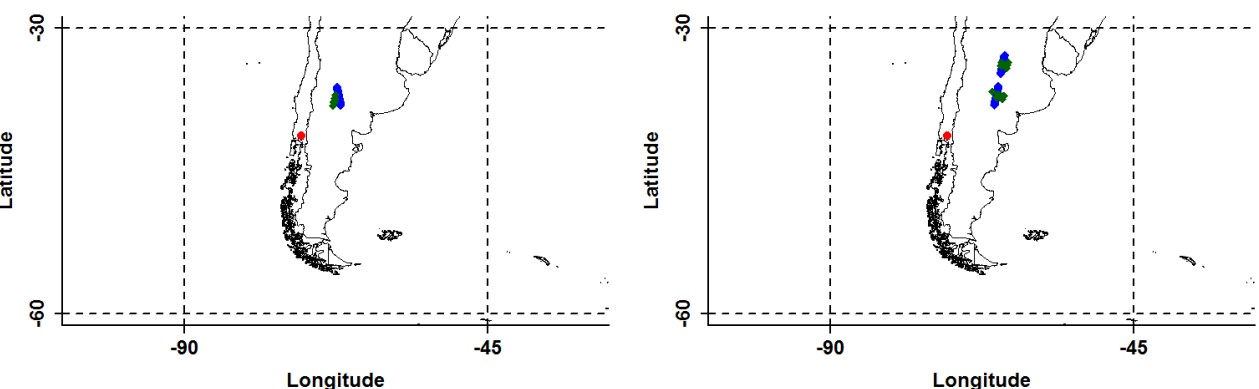

**Figure 9: The location of the volcano is shown in red. The location of the CALIOP and GOME-2B overpasses on the 23rd of April 2015 (left) and the 24th of April 2015 (right) are shown in blue and green respectively.**

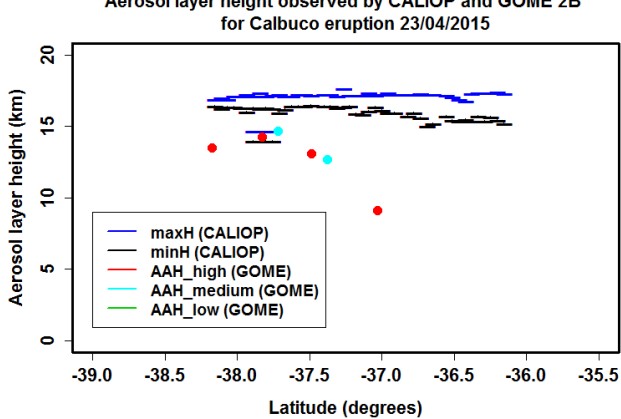

**Figure 10: The aerosol layer height detected by CALIOP is shown in blue (maximum layer height) and black (minimum layer height) as a function of latitude for the 23rd of April 2015. The GOME-2B AAH pixels located within 100 km of a CALIOP overpass pixel are presented in red (high reliability), cyan (medium reliability) and green (low reliability).**

The time difference between the CALIOP and GOME-2B overpass was between 4 and 5 hours and the closest GOME-2B pixel is located ~40 km away from a CALIOP pixel. Again, due to the inability of GOME-2 to observe layers higher than 15 km, the volcanic ash layer's height was underestimated by the AAH from GOME-2B.

The observed volcanic ash and dust layer height from CALIOP and the AAH detected by GOME-2B in function of latitude for the 24th of April 2015 are shown in Fig. 11. The time difference between both overpasses is around 8 hours and the closest GOME-2B pixel is located ~8 km away from a CALIOP pixel. The exact layer heights can be found in Table 9. On





this day, the high reliability AAH pixels of GOME-2B follow the height of the volcanic ash layer, whereas the medium reliability AAH matches the tropospheric dust and/or the polluted dust layer heights from CALIOP (Fig. 11).

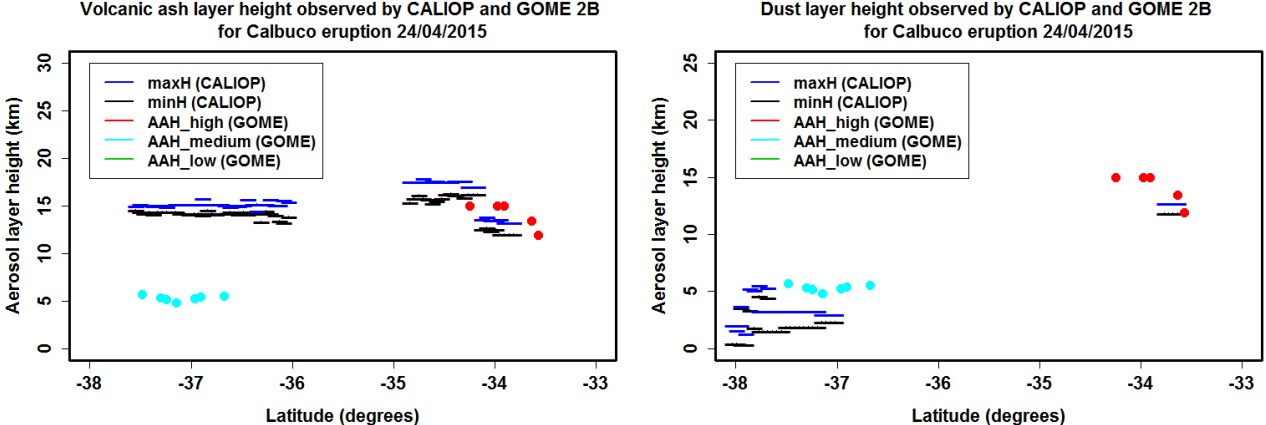

**Figure 11:** Detail of the volcanic ash (left) and dust (right) layer detected by CALIOP. The maximum and minimum layer heights are shown in blue and black respectively as a function of latitude for the 24th of April 2015. The GOME-2B AAH pixels located within 100 km of a CALIOP overpass pixel are presented in red (high reliability), cyan (medium reliability) and green (low reliability).

**Table 9:** Height of the aerosol layer detected by CALIOP for the different aerosol species compared to the AAH detected by GOME-2B for pixels located within 100 km distance for the 23rd and 24th of April (Calbuco eruption).

|  | 23rd of April 2015 | | 24th of April 2015 | |
|---|---|---|---|---|
|  | **CALIOP** | **AAH** | **CALIOP** | **AAH** |
| **Dust** | Not detected | Not detected | 0.2-12.7 km | 1.8-15 km |
| **Polluted continental** | Not detected | Not detected | 0.3-2.3 km | 11.9-15 km |
| **Clean continental** | Not detected | Not detected | 2.9-3.1 km | 2.2-3.1 km |
| **Polluted dust** | Not detected | Not detected | 0.2-5.5 km | 1.8-15 km |
| **Volcanic ash** | 13.2-18.6 km | 9.1-14.7 km | 12-17.9 km | 1.8-15 km |
| **Sulfate** | Not detected | Not detected | 12.6-21.7 km | 1.8-15 km |
| **Elevated smoke** | 15.4-16.9 km | 10.5-14.3 km | 12.7-17.5 km | 1.8-15 km |


## 3.2.2 Sarychev Peak

On the 14th of June 2009, a large eruption of the Sarychev Peak produced an ash plume that rose to an altitude of 12 km a.s.l. (Global Volcanism Program of the Smithsonian Institution, 2020). A large explosion the next day sent an ash release towards





an altitude of 8 km a.s.l. Data from the 14th and 16th of June 2009 from GOME-2A were studied. Figure 12 shows the
position of the GOME-2A and CALIOP overpasses near the Sarychev Peak volcano on the 14th and 16th of June 2009.

**GOME-2A**

On the 14th of June 2009, CALIOP detects clean marine, dust, polluted dust and dusty marine aerosol layers (Table 10). No
volcanic species have been observed. Fig. 13 shows the observed aerosol layer height from CALIOP and the AAH detected
by GOME-2A in function of latitude for the 14th of June 2009. The GOME-2A AAH slightly follows the height of the
CALIOP dust and dusty marine layers (Fig. 14). The time difference between GOME-2A and CALIOP overpasses is quite
large (15h) and the distance between the closest GOME-2A and CALIOP pixels is 11 km.

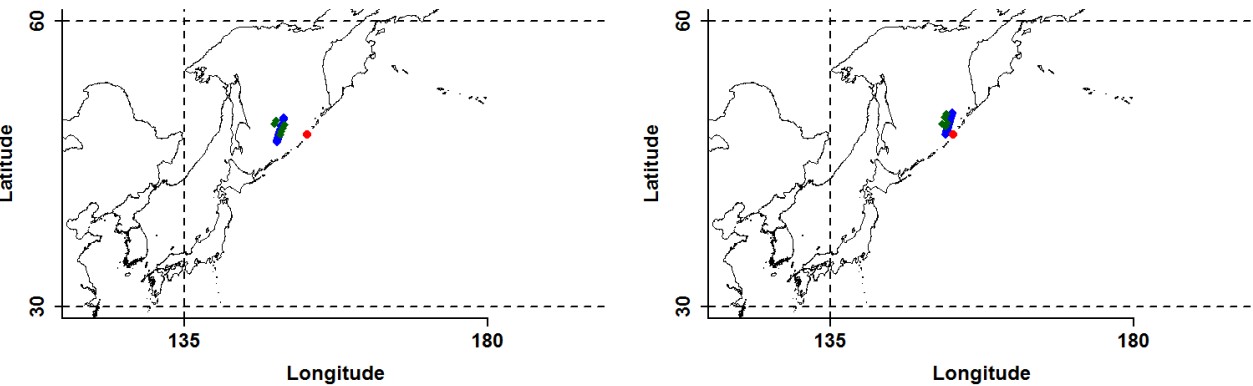

**Figure 12: The location of the volcano is shown in red. The location of the CALIOP and GOME-2A overpasses for the 14th (left)**
**and 16th (right) of June 2009 are shown in blue and green respectively.**

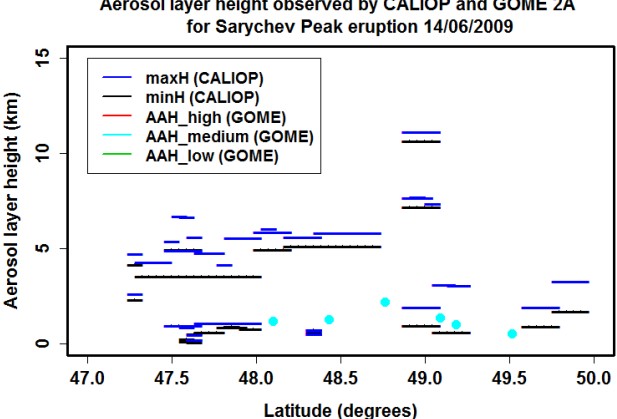

**Figure 13: The aerosol layer height detected by CALIOP is shown in blue (maximum layer height) and black (minimum layer**
**height) as a function of latitude for the 14th of June 2009. The GOME-2A AAH pixels located within 100 km of a CALIOP**
**overpass pixel are presented in red (high reliability), cyan (medium reliability) and green (low reliability).**


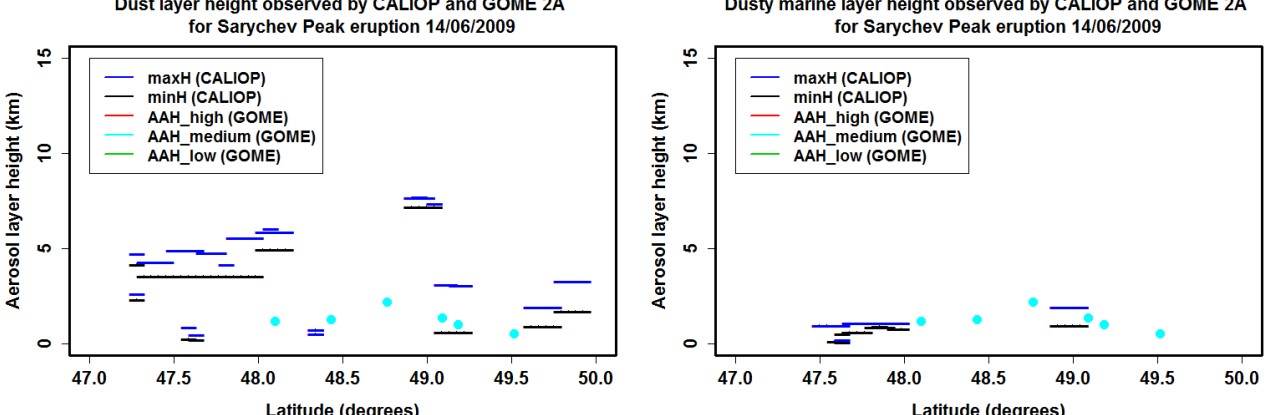

**Figure 14: Detail of the dust (left) and dusty marine (right) layer detected by CALIOP. The maximum and minimum layer heights are shown in blue and black respectively as a function of latitude for the 14th of June 2009. The GOME-2A AAH pixels located within 100 km of a CALIOP overpass pixel are presented in red (high reliability), cyan (medium reliability) and green (low reliability).**

On the 16[th] of June 2009, CALIOP detects dust, polluted dust, smoke, dusty marine and volcanic ash layers (Table 10). Even though a volcanic ash layer was observed by CALIOP at an altitude lower than 15 km (which should be detectable by GOME-2A), the AAH from GOME-2A does not match the height of this layer (Fig. 15). It seems that the AAH from GOME-2A agrees more with the height of the CALIOP dust and polluted dust layers (Fig. 15 and Table 10). The time difference between GOME-2A and CALIOP overpasses is large (15 h) and the distance between the closest GOME-2A and CALIOP pixels is 23 km.

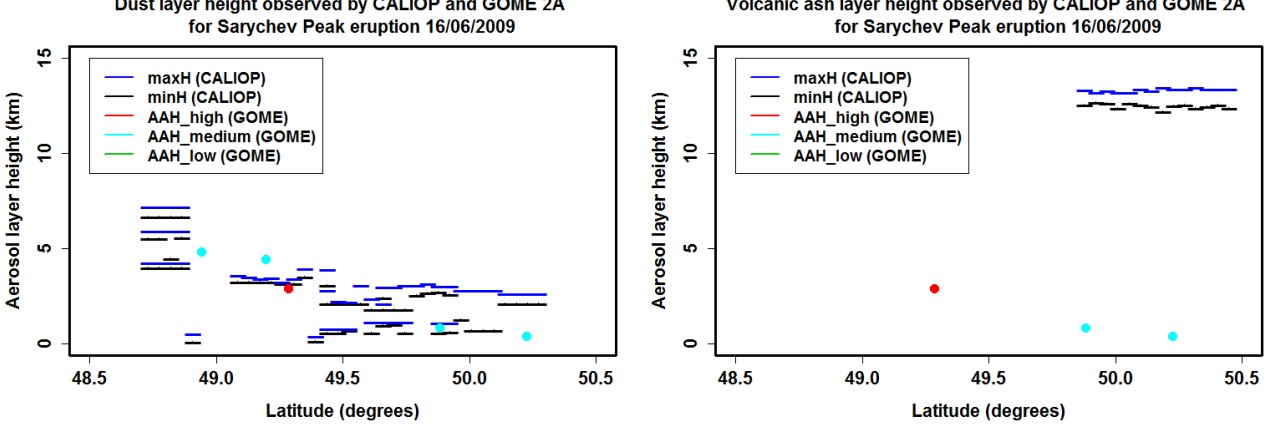

**Figure 15: Detail of the dust (left) and volcanic ash (right) layer detected by CALIOP. The maximum and minimum layer heights are shown in blue and black respectively as a function of latitude for the 16th of June 2009. The GOME-2A AAH pixels located within 100 km of a CALIOP overpass pixel are presented in red (high reliability), cyan (medium reliability) and green (low reliability).**



**Table 10: Height of the aerosol layer detected by CALIOP for the different aerosol species compared to the AAH detected by**
**GOME-2A for pixels located within 100 km distance for the 14th and 16th of June 2009 (Sarychev Peak eruption).**

|  | 14th of June 2009 | | 16th of June 2009 | |
|---|---|---|---|---|
|  | **CALIOP** | **AAH** | **CALIOP** | **AAH** |
| **Clean marine** | 0.6-0.7 km | 1.2-2.2 km | Not detected | Not detected |
| **Dust** | 0.2-7.7 km | 0.5-2.2 km | 0.1-7.2 km | 0.4-4.8 km |
| **Polluted dust** | 4.9-11.1 km | 0.5-2.2 km | 2.7-3.9 km | 0.4-4.8 km |
| **Smoke** | Not detected | Not detected | 6.3-6.9 km | 2.9-4.8 km |
| **Dusty marine** | 0.1-1.9 km | 0.5-2.2 km | 0.6-0.8 km | 2.9-4.8 km |
| **Clean marine** | 0.6-0.7 km | 1.2-2.2 km | Not detected | Not detected |
| **Volcanic ash** | Not detected | Not detected | 12.2-13.5 km | 0.4-2.9 km |

The above results show that for the Sarychev Peak eruption, the GOME-2A AAH more or less follows the height of the CALIOP (polluted) dust or dusty marine layers. Even though a volcanic ash layer is present below 15 km, within the detection limits of GOME-2, the instrument does not capture this layer. The time difference between the overpasses is large,
so it should be taken into account that the instruments are not looking at the same air mass.

## 4 Discussion

A few issues and complications encountered during the validation of GOME-2 AAH need to be addressed. First of all, it was challenging to find collocations both in space and time between GOME-2 and CALIOP overpasses. CALIOP has a very narrow footprint (100 m) and a far from global daily coverage, whereas GOME-2 has a near global and daily coverage with
ground pixels with a footprint of 80x40 km$^2$. Perfect collocations were therefore difficult to find, hence it was decided to set a threshold of 100 km for the maximum distance between the center of a GOME-2 pixel and the CALIOP coordinates. If we allowed for a larger distance threshold, the size of the dataset would increase, however it would also become more difficult to ensure that both satellites are looking at the 'same' air mass.

Currently, no threshold is fixed for the time difference between overpasses as this would limit our dataset even more.
However, by accepting all time differences, it might be possible that GOME-2 and CALIOP are not looking at the same air mass. Apart from finding collocations, not every volcanic eruption has GOME-2 and/or CALIOP overpasses within its plume and without trajectory modelling it is difficult to determine whether overpasses should observe volcanic species in their path. It was in some cases decided to look for overpasses further away from the actual volcano site, but again, it was challenging to state with absolute certainty that volcanic species should be present at that location.



Another factor limiting the dataset was that only cases with AAI higher than 4 were taken into account to ensure that the amount of absorbing aerosols is high enough (as discussed in the Tilstra et al., 2019b).

Difficulties also arose from the aerosol type classification used by CALIOP, which is partly based on the position of the layers in the atmosphere. CALIOP distinguishes between tropospheric (clean marine, dust, polluted continental, clean continental, dust, smoke and dusty marine types) and stratospheric (PSC aerosol, volcanic ash, sulfate and elevated smoke)

aerosol layers. It is known that due to this distinction based on altitude, volcanic aerosol types in the troposphere are sometimes misclassified as dust or polluted dust. As a consequence, not only the height of volcanic ash layers needs to be taken into account, but also the height of dust and polluted dust layers, while being not completely sure whether the dust layers are actually misclassified or not.

The performance of the AAH algorithm in representing the general aerosol layer height is far from optimal, as shown by the

results of the requirement analysis. The target threshold is only reached in 39%, 45% and 53% of the cases for GOME-2A, -2B and -2C respectively. The algorithm performs better in the troposphere, where the percentages increase to 52%, 47% and 57% and the mean errors improve (to $-0.2 \pm 3.6$ km, $-0.1 \pm 5.4$ km and $-0.8 \pm 3.8$ km for GOME-2A, -2B and -2C respectively for the minimum CALIOP layer height). It is shown that the algorithm performs better in retrieving the aerosol layer height for specific aerosol types (mainly the marine and continental aerosols). However, for the species that are of

interest in this study (i.e. volcanic ash in the stratosphere and (polluted) dust in the troposphere), GOME-2 AAH shows the biggest deviation to CALIOP aerosol layer heights. This indicates that GOME-2 is more sensitive to the signal of certain aerosol species.

It was already shown in Balis et al. (2016) that GOME-2A seems to strongly underestimate the ground based values. Here it was stated that it is highly likely that the large GOME-2 pixel size smooths out any small scale variability in the plume

height, which can be observed by the narrow measurement of CALIOP. Michailidis et al. (2020) validated the GOME-2 AAH by comparison with the aerosol layer height obtained from EARLINET stations and found a mean bias of $-0.18 \pm 1.68$ km for 172 screened collocations. This bias is smaller than the one we obtained. But we need to take into account first that the validation strategy is different. Michailidis et al. (2020) used ground based measurements for their validation, whereas we compare with satellite retrievals. Also, Michailidis et al. (2020) included AAH values for AAI between 2 and 4 in their

study, while this study is limited to AAI higher than 4. The allowed time and distance threshold is also different for both studies (100 km and no time limit (this study) vs 150 km distance and a 5 hour time limit (Michailidis et al., 2020)). This study focusses on the height of volcanic ash layers, whereas Michailidis et al. (2020) was focusing more on the general agreement in layer height for all aerosol types. So both studies have a different focus on the available dataset.

This study was based on a selection of case studies, representing volcanic events with a clear AAI signal (>4), which are important for aviation safety. The statistics presented in Sect. 3 are only valid for this dataset and cannot be extrapolated to the entire GOME-2 AAH dataset. At the moment, the product should be used carefully when assessing the height of volcanic layers and interpretations should only be made on a qualitative scale.



## 5 Conclusions


Within the framework of aviation safety, it is important to know the height of the volcanic ash layers. The GOME-2 AAH product was developed to provide a near global image of the height of absorbing aerosol layers. We presented the results of a validation exercise in which the GOME-2 AAH is validated using the aerosol layer height provided by CALIOP for a series of 15 confirmed volcanic eruptions. It is important to mention that only GOME-2 pixels for which the AAI was higher than 4

were taken into account. Also, a maximum difference of 100 km between the GOME-2 center pixel and the CALIOP overpass was allowed. No threshold was defined for the time difference between GOME-2 and CALIOP overpasses.

Overall, GOME-2A, GOME-2B and GOME-2C are able to represent the minimum CALIOP layer height with a mean error of -2.5 ± 5 km, -1.2 ± 5.9 km and -2 ± 5.8 km respectively. For the maximum CALIOP layer height, the mean errors are -3.3

± 5.1 km, -2.1 ± 5.9 km and -2.6 ± 5.9 km respectively. The high standard deviation is due to the inclusion of stratospheric aerosol species. If these stratospheric aerosol types are removed from the dataset, the errors become -0.2 ± 3.6 km, -0.1 ± 5.4 km and -0.8 ± 3.8 km for GOME-2A, GOME-2B and GOME-2C respectively for the minimum CALIOP layer height and -1.0 ± 3.6 km, -1.0 ± 5.4 km and -1.4 ± 3.9 km for GOME-2A, GOME-2B and GOME-2C respectively for the maximum CALIOP layer height. In the GOME-2 AAH product, reliability flags are used to define the confidence level of the AAH. It

would be expected that the high reliability AAH pixels have a better agreement with the CALIOP layer height, however, this was not always the case (e.g. for GOME-2A), which can be related to the difference in observed cases for the three sensors.

Some more conclusions could be drawn from looking at the volcanic case studies individually. It is obvious that the AAH product from GOME-2 does not work for elevated volcanic ash layers at altitudes higher than 15 km, due to the fact that the

FRESCO algorithm using the O2-A band is currently not sensitive for the signal at these altitudes. The Calbuco volcanic ash layer observed by CALIOP on the 23$^{rd}$ of April 2015, at altitudes above 15 km, could therefore not be captured by GOME-2A. On the other hand, GOME-2B was able to see, to some extent, the lower part of the absorbing volcanic aerosol layer on the 24$^{th}$ of April 2015. In the Calbuco case study, the AAH from GOME-2A pixels agreed with the height of layers classified by CALIOP as dust or polluted dust on the 24$^{th}$ of April 2015. It remains difficult to distinguish if these layers were volcanic

aerosol layers, actually misclassified by CALIOP, or if they were indeed dust or polluted dust. The study using the Sarychev Peak eruption data showed that, even though volcanic aerosols were present at altitudes below 15 km, GOME-2A was not able to pick up their signal.

When different types of aerosol layers are present, the AAH often coincides with one of the CALIOP species, but in most cases not with the layer classified as volcanic ash (as shown in Table 6 and Tables S1-S3 of the supplement). GOME-2 is in

some cases able to nicely capture the dust layer, but not always. The GOME-2A, -2B and -2C AAH pixels fall in the optimal threshold category in about 20%, 25% and 46% of all dust cases respectively. In conclusion, the GOME-2 AAH often





underestimates the height of volcanic layers and as a result, the current product should be considered with care when using it for aviation safety purposes. Nevertheless, taking these uncertainties into account, the product can be considered as an important added value for near-real time monitoring of volcanic ash layers.

## Author contribution

VDB: Writing – original draft preparation, Writing – review & editing, formal analysis, investigation

AM: Writing – review & editing

GT: Writing – review & editing

OT: Writing – review & editing

AD: Writing – review & editing, Supervision, project administration

## Competing interests

The authors declare that they have no conflict of interest.

## Acknowledgments

We thank KNMI for providing early access to the AC SAF AAH data for this validation study.

The CALIOP data were obtained from the NASA Langley Research Center Atmospheric Science Data Center.

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
