# Peer review of "Validation of the Absorbing Aerosol Height Product from GOME-2 using CALIOP Aerosol Layer Information"

_Atmospheric Measurement Techniques, 2020_

## Referee Comment (RC1) · Anonymous Referee #3 · 18 Jan 2021

In this paper, authors presented results from validation of GOME-2 Absorbing Aerosol Height (AAH) product against aerosol layer height from CALIOP Vertical Feature Mask (VFM) product for a selected suite of volcano cases. The objective of this paper is clear, which intends to present the performance evaluation of GOME-2 AAH product. However, there are several issues and challenges related to validation, which authors themselves also identified as well. First of all, AAH from GOME-2 represents centroid height of absorbing aerosol layer, while CALIOP identifies the height of each detected aerosol layer, so, it is not clear that how GOME-2 performs for a single layer vs. multi-layers of aerosols presented in the atmosphere. In another words, it is strongly suggested to authors to clarify how the maximum and minimum height is derived from CALIOP for

these two situations and how GOME-2 performs. Secondly, the authors' intention is to validate the performance of GOME-2 AAH for volcanic ash, which is the reason that the validation cases are specifically selected for volcanic eruptions. However, as noted by authors as well, CALIOP have troubles to give the correct type for volcanic ash, therefore, the analyses of GOME-2 performance for different aerosol types identified by CALIOP, seems to me, do not have any merit, and the type actually pre-defined due to case selection. Thirdly, the authors claimed that the performance of GOME-2 AAH does not have dependence on the distance and time in the matchups. This seems to be not convincing, since the analyses were performed when both factors are tangled together.

Some comments and questions are given as follows.

Major comments:

1. Figure 1 and subsequent figures shows one AAH values corresponds to multiples layer height values from CALIOP, it is evident that this is caused by the criteria used for matchups. So, instead of plot all points, why the authors cannot plot mean value and standard deviation from CALIOP for each GOME-2 AAH values? And also those large outliers are from matchups with a large distance difference or a large time difference? It is worth to investigate... 2. Analyses of Figure 2 and 3 seems have two factors tangled together. To clearly demonstrate the claim that the degree of agreement between GOME-2 and CALIOP does not have dependence on the difference in both space and time, the authors should bin one variable when analyzing the variability of the other variable. 3. It is understandable that matchup of GOME-2 with CALIOP is challenging, but the matchup criteria between GOME-2 and CALIOP might be too loose, which may lead to these large scatters in the results. Authors are encouraged to explore a relative tight criteria, it is ok to have less matchups, but do not want to include the matchups which really smear the results.

---

## Referee Comment (RC2) · Anonymous Referee #4 · 9 Mar 2021

This paper quantitatively validate the absorbing aerosol height (AAH) product from GOME-2 retrieved by the Fast Retrieval Scheme for Clouds from the Oxygen A band (FRESCO) algorithm for case studies of volcanic eruptions. The topic of this work is useful and provides the guide for the use of GOME-2 AAH product in volcano related research. However, the qualities of both the descriptions of the data and the presentations of results (figures and tables) have to be improved. In fact, previous studies about validation of passive satellite AAH retrieval with lidar observations always used lidar backscattter or aerosol extinction profiles (see references), but only CALIOP VFM product was used in this study. I don't think this is sufficient as a benchmark. If the authors have some specific reasons to use this product, these should be described. The

descriptions about the definition of AAH and aerosol profile assumption in FRESCO algorithm are also important to find an appropriate aerosol height benchmark from lidar measurements, but not involved in this study. The detailed comments are as follows.

**General comments**

1. Before the comparison of GOME-2 AAH and CALIOP data, the authors need to make the definition of GOME-2 AAH clearly, but I cannot find it. What is the aerosol/cloud profile assumption in the GOME-2 retrieval algorithm? For example, in EPIC/DSCOVR ALH retrieval, the aerosol profile is assumed to follow a quasi-Gaussian distribution (Xu et al. 2019, AMT; Xu et al. 2017, GRL), meanwhile in TROPOMI L2 ALH product, a uniform aerosol layer is defined and the middle layer is reported as ALH in the product (http://www.tropomi.eu/sites/default/files/files/publicSentinel-5P-TROPOMI-ATBD-Aerosol-Height.pdf). For different aerosol profile assumptions, the aerosol extinction shows different vertical distribution, which will affect the TOA reflectance oberved by the satellite. On the other hand, the definition of GOME-2 AAH also determines which height should be derived from CALIOP data to do comparison.

2. In fact, in many previous studies, the CALIOP extinction weighted aerosol height was always used as the validation of passive satellite retrieval, such as EPIC and TROPOMI (Xu et al. 2019, AMT; Nanda et al. 2020, AMT). Why did the authors choose the minimum and maximum layer from CALIOP data? How to define this minimum and maximum value? I think the CALIOP aerosol extinction product and backscatter data will be useful. From Fig. 1, the GOME-2 AAH seems not correlated with CALIOP data and indicates its bad agreement with CALIOP data and even meaningless.

[Figure]

3. When the authors did the comparison, was there any quality control method to be used to remove those unconfident AAH retrievals? If not, the invalid retrievals may reduce the correlation between GOME-2 AAH and CALIOP data in Fig.1. In fact, I do not quite understand the "accuracy requirements" in Table 1. More descriptions about the meaning of the numbers in this table is suggested to be added.

4. Most of the figures in this study is not impressive and the figures quality needs to be improved. For example, the data for different aerosol subtypes can be shown as different colored dots in Fig. 1a, so that the readers can easily find which aerosol type has better agreement and which is worse. Or the error in Fig. 4 can be expressed as errorbar instead of dots. The meaning of Fig. 1 and Fig. 4 have some overlaps and the authors should better organize them.

5. Actually, it is difficult to get aerosol type information when retrieve AAH only from O2 A band. Many algorithms only define a fixed aerosol model (optical properties) when retrieve AAH and this assumption will affect the AAH retrieval accuracy. What is the aerosol model used in the GOME-2 AAH retrieval algorithm? This may cause different accuracy when compared with different CALIOP subtypes height. If at one CALIOP footprint, there are several aerosol subtypes at different layers, how to do comparison with GOME-2 AAH?

6. Too many similar figures in Section 3.2. I suggest to re-organize them and provide more information in one figure. If the conclusions are similar for different cases, the figures do not have to be shown again.

[Figure]

**Specific comments**

1. Line 53-56: What is the relationship between the monitoring of surface thermal anomalies (from MODIS) and gas emissions (from OMPS or S5P) with plume extent detection? Maybe the plume chemical components and remote sensing characteristics could be mentioned here. I suggest to reword these two sentences.

2. I think the UV wavelength pair of GOME-2 AAI product used in the aerosol height retrieval could be mentioned at somewhere in the Introduction or Method, in case the readers are not famililar with this product.

3. Table 1: The meaning of threshold, target and optimal is suggested to be added in the table caption or nearby text.

4. I believe Table 2 is useless in this study.

5. Line 269: What does the "error" mean in Fig. 4? Does it represent the retrieval uncertainty? The authors should make it clear.

6. Figure 7: What does the GOME-2 data in this figure (and other similar figures hereafter) mean the closest pixel data or the mean value for all the pixels within 100 km to CALIOP footprint?

Xu, X., and Coauthors, 2017: Passive remote sensing of altitude and optical depth of dust plumes using the oxygen A and B bands: first results from EPIC/DSCOVR at Lagrange-1 point: Aerosol height retrieval from O2 A  B. Geophysical Research Letters, 44. Xu, X., and Coauthors, 2019: Detecting layer height of smoke aerosols over vegetated land and water surfaces via oxygen absorption bands: hourly results from EPIC/DSCOVR in deep space. Atmos. Meas. Tech., 12, 3269-3288. Nanda, S., M. de Graaf, J. P. Veefkind, M. Sneep, M. ter Linden, J. Sun, and P. F. Levelt, 2020:

A first comparison of TROPOMI aerosol layer height (ALH) to CALIOP data. Atmos. Meas. Tech., 13, 3043-3059. Sanders, A. F. J., de Haan, J. F., Sneep, M., Apituley, A., Stammes, P., Vieitez, M. O., Tilstra, L. G., Tuinder, O. N. E., Koning, C. E., and Veefkind, J. P.: Evaluation of the operational Aerosol Layer Height retrieval algorithm for Sentinel-5 Precursor: application to O2 A band observations from GOME-2A, Atmos. Meas. Tech., 8, 4947–4977, https://doi.org/10.5194/amt-8-4947-2015, 2015.

---

## Author Comment (AC1) · 15 Apr 2021

**Answer to anonymous referee #3:**

In this paper, authors presented results from validation of GOME-2 Absorbing Aerosol Height (AAH) product against aerosol layer height from CALIOP Vertical Feature Mask(VFM) product for a selected suite of volcano cases. The objective of this paper is clear, which intends to present the performance evaluation of GOME-2 AAH product. However, there are several issues and challenges related to validation, which authors themselves also identified as well.

First of all, AAH from GOME-2 represents centroid height of absorbing aerosol layer, while CALIOP identifies the height of each detected aerosol layer, so, it is not clear that how GOME-2 performs for a single layer vs. multi-layers of aerosols presented in the atmosphere. In another words, it is strongly suggested to authors to clarify how the maximum and minimum height is derived from CALIOP for two situations and how GOME-2 performs.

Added to the manuscript to clarify this:

"As long as the altitude difference between the different layers provided by the VFM product was less than 200 m, the different layers were considered as one big layer in this study. For these layers, the minimum (minC) and maximum (maxC) height was then determined. This was done to reduce the amount of CALIOP data to compare with GOME-2 overpasses."

Secondly, the authors' intention is to validate the performance of GOME-2 AAH for volcanic ash, which is the reason that the validation cases are specifically selected for volcanic eruptions. However, as noted by authors as well, CALIOP have troubles to give the correct type for volcanic ash, therefore, the analyses of GOME-2 performance for different aerosol types identified by CALIOP, seems to me, do not have any merit, and the type actually pre-defined due to case selection.

As stated, our study focuses on the quantitative validation of the GOME-2 AAH product for cases of volcanic eruptions. It is therefore natural that respective cases were selected. In section 2.2, the CALIOP Vertical Feature Mask (VFM) is described, with references describing the capabilities and uncertainties of the derivation of aerosol layers and types. Within these limitations, the aerosol type derivation by CALIOP can be regarded as correct. Although our study focuses on GOME-2 AAH and volcanic ash, we think that it is worthwhile to present also the performance for the AAH not only when looking at all heights for all aerosol types but also for the different types. It is already mentioned that these analyses have to be taken with caution (cf. lines 341-342). We will add, however, some text to state this more clearly:
in line 297:
"Note, that the results for different aerosol types have to be analysed with caution, as the data were retrieved specifically for periods of volcanic eruptions".
And also in the supplement, in line 8:
"Note, that the results for different aerosol types have to be analysed with caution, as the data were retrieved for periods of volcanic eruptions."

Thirdly, the authors claimed that the performance of GOME-2 AAH does not have dependence on the distance and time in the matchups. This seems to be not convincing, since the analyses were performed when both factors are tangled together.

Please see our answer to this point further below (changes to figures 2 and 3).

Some comments and questions are given as follows.
Major comments:
1. Figure 1 and subsequent figures shows one AAH values corresponds to multiples layer height values from CALIOP, it is evident that this is caused by the criteria used for matchups. So, instead of plot all points, why the authors cannot plot mean value and standard deviation from CALIOP for each GOME-2 AAH values? And also those large outliers are from matchups with a large distance difference or a large time difference? It is worth to investigate...

Figure 1 has been adjusted according to the reviewer's comments and now shows for each GOME-2 AAH value the mean and standard deviation of the corresponding minimum CALIOP layer heights.

The outliers are discussed in the text (lines 281-286):
"In Fig. 1 there are points for which the average CALIOP minimum layer height is higher than 12 km and the corresponding GOME-2 AAH is much lower (< 3 km for GOME-2A and < 9 km for GOME-2B). For GOME-2A, most of the corresponding individual CALIOP pixels (85 %) were classified as volcanic ash, sulfate or elevated smoke layers and are classified by GOME-2 as pixels with high reliability. For GOME-2B however, only 28% of the corresponding individual CALIOP pixels were classified as stratospheric aerosol species but 95% of the GOME-2B pixels have a medium or low reliability level."

**Changes to the manuscript:**

New figure 1:

[Figure]

Fig.1: Requirement plots for GOME-2A (upper left), GOME-2B (upper right) and GOME-2C (lower middle). The red, green and blue line represent the threshold, target and optimal requirement lines. For each GOME AAH, the corresponding mean minimum CALIOP layer height and the standard deviation are shown.

2. Analyses of Figure 2 and 3 seems have two factors tangled together. To clearly demonstrate the claim that the degree of agreement between GOME-2 and CALIOP does not have dependence on the difference in both space and time, the authors should bin one variable when analyzing the variability of the other variable.

We created a new figure that combines the information from both figures 2 and 3, using colors to represent the different time difference bins.
As the results do not change, the adaptions in the text (paragraph lines 287-295) are limited to referring to Fig. 2 only and to delete Fig. 3 references.

**Changes to the manuscript:**

New figure 2: (+ removed figure 3)

[Figure]

Fig. 2: Difference between GOME-2 AAH and the minimum CALIOP layer height in function of the distance between the GOME-2 and CALIOP pixel. The different colors represent different classes of time differences between the GOME-2 and CALIOP overpasses. The upper left plot shows the results for GOME-2A, the upper right plot shows the results for GOME-2B and the lower middle plot shows the results for GOME-2C.

3. It is understandable that matchup of GOME-2 with CALIOP is challenging, but the matchup criteria between GOME-2 and CALIOP might be too loose, which may lead to these large scatters in the results. Authors are encouraged to explore a relative tight criteria, it is ok to have less matchups, but do not want to include the matchups which really smear the results.

We studied the impact of using stricter criteria both for the maximum distance and for the time difference between GOME-2 and CALIOP overpasses separately.

First, we lowered the maximum distance from 100 km to 50 km. This reduced the dataset significantly, as only about 20%, 23% and 12% of the original data remained for GOME-2A, -2B and -2C respectively. When looking at the amount of data corresponding to the accuracy requirements, we found that by applying this tighter distance threshold, the amount of data in the threshold, target and optimal categories was reduced. In the scatterplot, there still remained a cloud of points with low AAH and high CALIOP minimum layer heights. This cloud represented 24%, 21% and 20% of the remaining data set for GOME-2A, -2B and -2C respectively.

Then, we kept the original distance limit of 100 km but we applied a threshold on the time between overpasses and only selected overpasses within 6h of each other. This also reduced the dataset: about 50%, 40% and 3% of the original data remained for GOME-2A, -2B and -2C respectively. For GOME-2A and -2B, the amount of data agreeing with the threshold and target requirements increased. For GOME-2C, too little data remained and the amount of data in the threshold, target and optimal requirements decreased to 23%, 8% and 0% respectively. Also for the reduced time difference, there still remained a cloud of points with low AAH and high CALIOP minimum layer heights. This cloud represented 21% and 20% of the remaining data set for GOME-2A and -2B respectively.

We decided to keep the original analysis with no limit on the time between GOME-2 and CALIOP overpasses and with a limit of 100 km for the maximum distance between overpasses. Our data set at the moment is too reduced by putting more strict thresholds in place and the improvement in the statistics is too small to make the change worthwhile. In the future when more data can be included in our analysis, we will revisit the option to impose stricter thresholds.

---

## Author Comment (AC2) · 15 Apr 2021

**Answer to anonymous referee #4:**

This paper quantitatively validate the absorbing aerosol height (AAH) product from GOME-2 retrieved by the Fast Retrieval Scheme for Clouds from the Oxygen A band (FRESCO) algorithm for case studies of volcanic eruptions. The topic of this work is useful and provides the guide for the use of GOME-2 AAH product in volcano related research. However, the qualities of both the descriptions of the data and the presentations of results (figures and tables) have to be improved. In fact, previous studies about validation of passive satellite AAH retrieval with lidar observations always used lidar backscattter or aerosol extinction profiles (see references), but only CALIOP VFM product was used in this study. I don't think this is sufficient as a benchmark. If the authors have some specific reasons to use this product, these should be described. The descriptions about the definition of AAH and aerosol profile assumption in FRESCO algorithm are also important to find an appropriate aerosol height benchmark from lidar measurements, but not involved in this study. The detailed comments are as follows.

**General comments**

1. Before the comparison of GOME-2 AAH and CALIOP data, the authors need to make the definition of GOME-2 AAH clearly, but I cannot find it. What is the aerosol/cloud profile assumption in the GOME-2 retrieval algorithm? For example, in EPIC/DSCOVR ALH retrieval, the aerosol pro-file is assumed to follow a quasi-Gaussian distribution (Xu et al.2019,AMT; Xu et al.2017, GRL), meanwhile in TROPOMI L2 ALH product, a uniform aerosol layer is defined and the middle layer is reported as ALH in the product (http://www.tropomi.eu/sites/default/files/files/publicSentinel-5P-TROPOMI-ATBD-Aerosol-Height.pdf). For different aerosol profile assumptions, the aerosol extinction shows different vertical distribution, which will affect the TOA reflectance observed by the satellite. On the other hand, the definition of GOME-2 AAH also determines which height should be derived from CALIOP data to do comparison.

To specifically answer your question about the aerosol/cloud profile used:
The retrieval of the AAH is based on the FRESCO method, where no specific aerosol layer/profile was introduced. Instead a Lambertian layer has been inserted in the model.
The manuscript will be updated to include more background information about the AAH. More detailed information will be extracted from Wang et al. 2012 and Tilstra et al. 2019a.

**Changes to the manuscript:**

Added the following text to section 2.1.2:

"The AAH algorithm is designed to handle GOME-2 level-1b Product Dissemination Units (PDUs) (Tilstra et al., 2019a). First, however, the associated AAI level-2 PDU is opened and each observation in it is examined. Observations with solar zenith angles larger than 85 degrees, observations in sun glint geometries and observations known to be affected by a solar eclipse event are skipped. Whenever the AAI is below 4, the amount of absorbing aerosol is too small to result in a reliable retrieval of the AAH. In this case the observation is skipped and the AAH is returned as not determined. If the AAI value is higher than 4, then the algorithm will try to retrieve aerosol layer height in the following way.
In order to simulate the reflectance spectrum of a partly cloudy pixel inside and outside the $O_2$ A band, a simple atmospheric model is used, in which the atmosphere above the ground surface (for the cloud free part of the pixel) or cloud (for the cloudy part of the pixel) is treated as an absorbing (due to oxygen) and purely Rayleigh scattering medium (Wang et al. 2012). Reflection occurs only at the surface and the cloud top. Surface and cloud are assumed to be Lambertian reflectors.
The reflectance $R_{sim}(\lambda, \theta, \theta_0, \phi-\phi_0)$ at wavelength $\lambda$, viewing zenith angle $\theta$, solar zenith angle $\theta_0$, and relative azimuth angle $\phi-\phi_0$ is then given by

$$R_{sim} = cT_c(z_c)A_c + cR_c(z_c) + (1-c)T_s(z_s)A_s + (1-c)R_s(z_s). \qquad (1)$$

If c = 1, the surface related terms vanish and the above equation is simplified to
$$R_{sim} = T_c(z_{sc})A_c + R_c(z_{sc}). \qquad\qquad\qquad (2)$$

Note that the wavelength and directional dependencies are omitted in equations (1) and (2) for $R_{sim}$, $T_c$, $R_c$, $T_s$, and $R_s$. In the above equations, c is the effective aerosol/cloud cover fraction at the O2-A band, $A_c$ is the albedo of the aerosol/cloud layer, $A_s$ is the surface albedo, and $A_{sc}$ is the scene albedo. The terms $T(\lambda, z_s, \theta, \theta_0)$, $T(\lambda, z_c, \theta, \theta_0)$, and $T(\lambda, z_{sc}, \theta, \theta_0)$ are the direct atmospheric transmittances for light entering the atmosphere from the solar direction, propagating down to different levels characterized by surface height $z_s$, aerosol/cloud height $z_c$, and scene height $z_{sc}$, respectively, then propagating to the top of the atmosphere (TOA) in the direction of the satellite. Oxygen absorption and single Rayleigh scattering are both taken into account in the light paths for the transmittances and in the single Rayleigh scattering reflectances above the aerosol/cloud layer ($R_c$) and the surface ($R_s$), respectively, in the way as described in, for instance, Wang et al. (2008). The transmittances and reflectances are pre-calculated and stored in a look-up table.

The GOME-2 AAH algorithm essentially retrieves the aerosol layer height using two approaches. In the first approach, equation (1) is used, and the aerosol/cloud layer height $z_c$ is retrieved along with effective aerosol/cloud cover fraction c. The aerosol layer albedo $A_c$ is set to a fixed value of 0.8, which is an appropriate value for clouds (Koelemeijer et al., 2001; Wang et al., 2008) and also a functional value for thick aerosol layers (Wang et al., 2012). The parameters retrieved this way are in fact identical to the parameters retrieved by the FRESCO+ cloud information retrieval. In the second approach, the scene albedo $A_{sc}$ and scene height $z_{sc}$ are derived using equation (2), i.e., by assuming the aerosol/cloud fraction c to be equal to one (Koelemeijer et al., 2001; Stammes et al.,2008; Wang et al., 2008). Large aerosol plumes often cover several GOME-2 pixels. Thus, it seems reasonable to assume an aerosol/cloud cover fraction of one in these situations."

+ references added to reference list:

Koelemeijer, R. B. A., Stammes, P., Hovenier, J.W. and de Haan, J.F.: A fast method for retrieval of cloud parameters using oxygen a band measurements from the Global Ozone Monitoring Experiment, J. Geophys. Res., 106(D4), 3475–3490, doi:10.1029/2000JD900657, 2001.

Stammes, P., Sneep, M., de Haan, J.F., Veefkind, J.P., Wang, P. and Levelt, P.F.: Effective cloud fractions from the Ozone Monitoring Instrument: Theoretical framework and validation, J. Geophys. Res., 113, D16S38, doi:10.1029/2007JD008820, 2008.

2. In fact, in many previous studies, the CALIOP extinction weighted aerosol height was always used as the validation of passive satellite retrieval, such as EPIC and TROPOMI (Xu et al. 2019, AMT; Nanda et al. 2020, AMT). Why did the authors choose the minimum and maximum layer from CALIOP data? How to define this minimum and maximum value? I think the CALIOP aerosol extinction product and backscatter data will be useful. From Fig. 1, the GOME-2 AAH seems not correlated with CALIOP data and indicates its bad agreement with CALIOP data and even meaningless.

None of the authors have the knowhow or software infrastructure to work with the CALIOP extinction weighted aerosol height. We decided to use the Vertical Feature Mask product developed by NASA (Liu et al., 2005, Omar et al., 2009, Kim et al., 2018) which transforms the original total backscatter signal from CALIOP to a product representing the height of different aerosol layers and types:

From Nowottnick et al. 2015:
*"CALIOP provides daytime and nighttime attenuated backscatter profiles at 532 and 1064 nm, as well as information about depolarization at 532 nm. This information is first used to discriminate cloud and aerosol layers (Liu et al.,2005). Aerosol layers are then fed through a vertical feature mask (VFM) algorithm that assigns aerosol type classifications based on aerosol geographic location, the underlying surface type, observed aerosol altitude, attenuated backscatter, and depolarization ratio. The practical*

*application of the CALIOP VFM is to assign an appropriate lidar ratio for each detected aerosol layer in order to compute aerosol extinction profiles from the attenuated backscatter signals, extinction being more directly comparable to model fields than attenuated backscatter (Omar et al., 2009). By itself, though, the VFM also provides a unique tool for directly exploring the vertical distribution of cloud and aerosol layers, as well as aerosol composition."*

We trust the quality of this product as it was developed by members of the CALIOP team. Several papers have also used this product (e.g. Adams et al., 2012, Burton et al., 2013, Chen et al., 2012, Hagihara et al., 2010, Mielonen et al., 2009, Nowottnick et al., 2015, Tesche et al., 2013 and Yoshida et al., 2010). We believe that developing a method using the extinction signal is out of the scope of this article, as we do not want to focus on the quality and retrieval of CALIOP data but instead on the new GOME-2 AAH product.

References CALIOP VFM articles:
Adams, A. M., Prospero, J. M., and Zhang, C.: CALIPSO-Derived Three-Dimensional Structure of Aerosol over the Atlantic Basin and Adjacent Continents, J. Climate, 25, 6862–6879, 2012.

Burton, S. P., Ferrare, R. A., Vaughan, M. A., Omar, A. H., Rogers, R. R., Hostetler, C. A., and Hair, J. W.: Aerosol classification from airborne HSRL and comparisons with the CALIPSO vertical feature mask, Atmos. Meas. Tech., 6, 1397–1412, doi:10.5194/amt-6-1397-2013, 2013.

Chen, Z., Torres, O., McCormick, M. P., Smith, W., and Ahn, C.: Comparative study of aerosol and cloud detected by CALIPSO and OMI, Atmos. Environ., 51, 187–195, 2012.

Hagihara, Y., Okamoto, H., and Yoshida, R.: Development of a combined CloudSat-CALIPSO cloud mask to show global cloud distribution, J. Geophys. Res., 115, D00H33,doi:10.1029/2009JD012344, 2010.

Kim, M.-H., Omar, A. H., Tackett, J. L., Vaughan, M. A., Winker, D. M., Trepte, C. R., Hu, Y., Liu, Z., Poole, L. R., Pitts, M. C., Kar, J. and Magill, B. E.: The CALIPSO version 4 automated aerosol classification and lidar ratio selection algorithm, Atmos. Meas. Tech., 11, 6107-6135, doi: 10.5194/amt-11-6107-2018, 2018.

Liu, Z., Omar, A. H., Hu, Y., Vaughan, M. A., Winker, D. M., Poole, L. R., and Kovacs, T. A.: CALIOP algorithm theoretical basis document, Part 3: Scene classification algorithms, NASA-CNES document PC-SCI-203, 2005.
Mielonen, T., Arola, A., Komppula, M., Kukkonen, J., Koskinen, J., de Leeuw, G., and Lehtinen, K. E. J.: Comparison of CALIOP level 2 aerosol subtypes to aerosol types derived from AERONET inversion data, Geophys. Res. Lett., 36, L18804, doi:10.1029/2009GL039609, 2009.

Nowottnick, E.P., Colarco, P.R., Welton, E.J and da Silva, A.: Use of the CALIOP vertical feature mask for evaluating global aerosol models, Atmos. Meas. Tech., 8, 3647-3669, doi: 10.5194/amt-8-3647-2015, 2015.

Omar, A. H., Winker, D. M., Kittaka, C., Vaughan, M. A., Liu, Z., Hu, Y., and Hostetler, C. A.: The CALIPSO automated aerosol classification and lidar ratio selection algorithm, J. At-mos. Ocean. Tech., 26, 1994–2014, 2009.

Tesche, M., Wandinger, U., Ansmann, A., Althausen, D., Müller, D., and Omar, A. H.: Ground-based validation of CALIPSO observations of dust and smoke in the Cape Verde region, J. Geo-phys. Res. Atmos., 118, 2889–2902, doi:10.1002/jgrd.50248,2013.

Vaughan, M., Pitts, M., Trepte, C., Winker, D., Detweiler, P., Garnier, A., Getzewich, B., Hunt, W., Lambeth, J., Lee, K.-P., Lucker, P., Murray, T., Rodier, S., Tremas, T., Bazureau, A. and Pelon, J.: Cloud-Aerosol LIDAR Infrared Pathfinder Satellite Observations – Data Management System – Data Products Catalog, Release 4.70, Document No: PC-SCI-503, 2019.

Vaughan, M. A., Winker, D. M., and Powell, K. A.: CALIOP algorithm theoretical basis document, part 2: Feature detection and layer properties algorithms, Rep. PC-SCI, 202, 87, 2005.

Yoshida, R., Okamoto, H., Hagihara, Y., and Ishimoto, H.: Global analysis of cloud phase and ice crystal orientation from Cloud Aerosol Lidar and Infrared Pathfinder Satellite Observation (CALIPSO) data using attenuated backscattering and depolarization ratio, J. Geophys. Res., 115, D00H32,doi:10.1029/2009JD012334, 2010.

**Explanation min/max CALIOP height:** The CALIOP VFM product provides the latitude, longitude, altitude and aerosol classification of all individual layers. As long as the altitude difference between the different layers was less than 200 m, the different layers were considered as one big layer. For this layer, the minimum and maximum height was then determined. This was done to reduce the amount of CALIOP data to compare with GOME-2 overpasses.

**Changes to the manuscript:**

Added to section 2.2:

"As long as the altitude difference between the different layers provided by the VFM product was less than 200 m, the different layers were considered as one big layer in this study. For these layers, the minimum (minC) and maximum (maxC) height was then determined. This was done to reduce the amount of CALIOP data to compare with GOME-2 overpasses."

3. When the authors did the comparison, was there any quality control method to be used to remove those unconfident AAH retrievals? If not, the invalid retrievals may reduce the correlation between GOME-2 AAH and CALIOP data in Fig.1.In fact, I do not quite understand the "accuracy requirements" in Table 1. More descriptions about the meaning of the numbers in this table is suggested to be added.

The only quality control method used to remove unconfident AAH retrievals is the selection of AAH data for which the AAI is higher than 4 to ensure that we only investigate data for which the aerosol absorption was high enough. Also, observations with solar zenith angles larger than 85 degrees, observations in sun glint geometries and observations known to be affected by a solar eclipse event are skipped (Tilstra et al., 2019).
The low, medium and high reliability labels are given based on whether the cloud height or scene height is used to determine the AAH. This is explained in section 2.1.2 in the manuscript.
The accuracy requirements presented in table 1 are developed and defined within the steering group of EUMETSAT AC SAF. The numbers they defined are obtained after an extensive procedure. We added this information to the text.

**Changes to the manuscript:**

Added to section 2.1.2:

"Observations with solar zenith angles larger than 85 degrees, observations in sun glint geometries and observations known to be affected by a solar eclipse event are skipped. Whenever the AAI is below 4, the amount of absorbing aerosol is too small to result in a reliable retrieval of the AAH."

+ "The accuracy requirements for the AAH product, **as defined by the AC SAF steering committee** in the Product Requirements Document (Hovila et al., 2019), can be found in Table 1."

4. Most of the figures in this study is not impressive and the figures quality needs to be improved. For example, the data for different aerosol subtypes can be shown as different colored dots in Fig. 1a, so that the readers can easily find which aerosol type has better agreement and which is worse. Or the error in Fig. 4 can be expressed as error bar instead of dots. The meaning of Fig. 1 and Fig. 4 have some overlaps and the authors should better organize them.

We decided to change Fig. 1 by plotting for each GOME-2 AAH value the mean and standard deviation of the corresponding CALIOP minimum layer heights (as suggested by reviewer 1).
The contents of Figure 2 and 3 are now merged in one new figure 2 where the difference between the AAH and the minimum CALIOP layer height in function of distance is shown for different bins of time difference (represented by the different colors).
We agree that Figure 4 does not add much information, so we decided to remove it from the manuscript.
We improved the quality of the figures.

**Changes to the manuscript:**

New figure 1 + new figure 2 + removed figures 3 and 4.

[Figure]

Fig.1: Requirement plots for GOME-2A (upper left), GOME-2B (upper right) and GOME-2C (lower middle). The red, green and blue line represent the threshold, target and optimal requirement lines. For each GOME AAH, the corresponding mean minimum CALIOP layer height and the standard deviation are shown.

[Figure]

Fig. 2: Difference between GOME-2 AAH and the minimum CALIOP layer height in function of the distance between the GOME-2 and CALIOP pixel. The different colors represent different classes of time differences between the GOME-2 and CALIOP overpasses. The upper left plot shows the results for GOME-2A, the upper right plot shows the results for GOME-2B and the lower middle plot shows the results for GOME-2C.

5. Actually, it is difficult to get aerosol type information when retrieve AAH only from O2 A band. Many algorithms only define a fixed aerosol model (optical properties) when retrieve AAH and this assumption will affect the AAH retrieval accuracy. What is the aerosol model used in the GOME-2 AAH retrieval algorithm? This may cause different accuracy when compared with different CALIOP subtypes height. If at one CALIOP footprint, there are several aerosol subtypes at different layers, how to do comparison with GOME-2 AAH?

The AAH retrieval method (using the FRESCO algorithm) does not use a specified aerosol model. A general Lambertian layer is inserted in the model (see also answer to general comment 1).
For each GOME pixel, all individual CALIOP layers located at a maximum distance of 100 km are taken into account for the comparison between the GOME-2A AAH and the CALIOP layer height. This indeed means that for a certain CALIOP footprint, multiple aerosol layers of the same or different aerosol types may be present. In that case all these layers will be individually compared to the GOME-2 AAH.

6. Too many similar figures in Section 3.2. I suggest to re-organize them and provide more information in one figure. If the conclusions are similar for different cases, the figures do not have to be shown again.

Several figures have been moved to the supplement.

**Specific comments**

1. Line 53-56: What is the relationship between the monitoring of surface thermal anomalies (from MODIS) and gas emissions (from OMPS or S5P) with plume extent detection? Maybe the plume chemical components and remote sensing characteristics could be mentioned here. I suggest to reword these two sentences.

The authors were giving general examples of different satellite instruments that can monitor the presence of volcanic layers. We don't want to go into the details of the different techniques as this is not the scope of this paper. We will rephrase these sentences.

**Changes to the manuscript:**

"Volcanic plumes can also be monitored by the Ozone Monitoring Instrument (OMI), Ozone Mapping and Profiler Suite (OMPS) and Sentinel-5p (S5p) (they are able to provide gaseous emission mapping) and by MODerate-resolution Imaging Spectroradiometer (MODIS) (through mapping of surface thermal anomalies)."

2. I think the UV wavelength pair of GOME-2 AAI product used in the aerosol height retrieval could be mentioned at somewhere in the Introduction or Method, in case the readers are not familiar with this product.

The wavelength pair used for the AAI product is the 340-380 nm pair. This information was added to the manuscript in section 2.1.2.

**Changes to manuscript:**

"This product builds on a previously developed product, the AAI **(derived from the 340-380 nm wavelength pair; Tuinder et al., 2019)**…"

3. Table 1: The meaning of threshold, target and optimal is suggested to be added in the table caption or nearby text.

The accuracy requirements presented in table 1 are developed and defined within the steering committee of EUMETSAT AC SAF. The numbers they defined are obtained after an extensive procedure. To satisfy the threshold, target and optimal requirement, the AAH should be within respectively 3, 2 and 1 km (4, 3 and 2 km) of the actual height of the absorbing aerosol layer for layers lower (higher) than 10 km altitude.

**Changes to the manuscript:**

"The accuracy requirements for the AAH product, **as defined by the AC SAF steering committee** in the Product Requirements Document (Hovila et al., 2019), can be found in Table 1."

+ adjusted caption of table 1:

Table 1. Accuracy requirements defined **by the AC SAF steering group** for the AAH product (from Hovila et al., 2019).

4. I believe Table 2 is useless in this study.

We moved this table to the supplement

5. Line 269: What does the "error" mean in Fig. 4? Does it represent the retrieval uncertainty? The authors should make it clear.

The error of the AAH is provided by KNMI. It is calculated using the error propagation theory where the uncertainty in the measured spectral (ir)radiances transforms into an error in the calculated reflectances and hence in all the derived products, including the AAH. This error does not contain real information on the quality of the AAH.

**Changes to manuscript:**

Figure 4 was removed + text added to section 3.1:

"The GOME-2 AAH product also provides a calculated error for each AAH value. This error is calculated using the error propagation theory where the uncertainty in the measured spectral (ir)radiances transforms into an error in the calculated reflectances and hence in all the derived products, including the AAH. This error does not contain real information on the quality of the AAH."

6. Figure 7: What does the GOME-2 data in this figure (and other similar figures hereafter) mean the closest pixel data or the mean value for all the pixels within100 km to CALIOP footprint?

In this type of figures, all individual GOME-2 pixels around the location of the volcano are shown together with the individual CALIOP overpasses that are located at a maximum distance of 100 km of a GOME-2 overpass.